# Understanding American premium chocolate consumer perception of craft chocolate and desirable product attributes using focus groups and projective mapping

Allison L. Brown[1], Alyssa J. Bakke[1,2], Helene Hopfer[1,2]*

1 Department of Food Science, The Pennsylvania State University, University Park, Pennsylvania, United States of America, 2 Sensory Evaluation Center, The Pennsylvania State University, University Park, Pennsylvania, United States of America

* hopfer@psu.edu

## 

**Data Availability Statement:** All relevant data underlying the results presented in the study that can be publicly displayed are available within the

## Abstract

Craft chocolate is a relatively new and fast-growing segment of the American chocolate market. To understand American premium chocolate consumer perception of craft chocolate and desirable chocolate product attributes, we conducted a mixed-methods study using focus groups and projective mapping. Projective mapping revealed that participants segmented products in terms of quality based upon usage occasion rather than cost or other factors. We found that American premium chocolate consumers use search attributes such as segmentation, price, availability, and packaging as quality determinants. Additionally, they desire credence attributes that convey trust through, for example, the presence or absence of sustainability certifications, or a semblance of meaning. Premium chocolate consumers seek out experience attributes such as utility and/or joy, which are achieved by purchasing a chocolate product as a gift, for its nostalgic purposes, or for desired post-ingestive effects. We propose a Desirable Chocolate Attribute Concept Map to explain our findings.

## Introduction

"Craft" or "bean-to-bar" chocolate has experienced prolific growth in the United States chocolate market in the past twenty-plus years. The American craft chocolate industry is said to have begun in 1996 with Scharffen Berger Chocolate Maker who coined the term "bean-to-bar" while making chocolate from cocoa beans in their Berkeley, California garage [1–3]. As of 2015, Leissle reported that there were 129 craft chocolate makers in the United States [4] and by 2016, Woolley et al. reported that this number had grown to 177 makers [5]. In 2018, the Fine Chocolate Industry Association (FCIA), an industry group, stated that there were over 300 craft chocolate makers, most of which are located in the United States [6]. There is no official definition for craft or bean-to-bar chocolate, though several definitions have been proposed [1, 2]. In 2008, the now-defunct group, Craft Chocolate Makers of America, defined craft chocolate as "made from scratch by an independent, small company (one that uses

paper, its Supporting Information files, or at doi.org/10.26207/a863-pp02. The Pennsylvania State University Institutional Review Board reviewed and approved the study protocol (number 6654). This protocol limits the use and public sharing of the transcripts for any future research due to the presence of sensitive identifying information. Field data access queries may be directed to the Research Data Management Services at The Pennsylvania State University Libraries (contact via L-DATA-MGMT@lists.psu.edu).

**Funding:** This work was supported by the United States Department of Agriculture (USDA) National Institute of Food and Agriculture (NIFA) Federal Appropriations under Project PEN04624 and Accession number 1013412, awarded to H.H. (https://nifa.usda.gov/grants). The funders had no role in study design, data collection and analysis, decision to publish, or preparation of the manuscript. This research did not receive any industry sponsorship. The products used were selected at the discretion of the authors.

**Competing interests:** ALB, AJB, and HH are all employed by Penn State. ALB is a former employee of Ghent University and was an intern at Mars Netherlands. AJB is a former employee of Land O'Lakes and has received consulting fees from Eight Oaks Distillery and Giant Eagle. HH is a former employee of UC Davis and HM.Clause and has received consulting fees from Henkel Adhesive Technology. HH is an associate editor and member of the publications committee at the American Society of Enology & Viticulture (ASEV). ALB receives funding from USDA-NIFA. AJB receives or has received funding from PA Dept. of Agriculture and the National Sugar Association. HH receives or has received funding from Penn State, University of Alabama, USDA-NIFA, USDAFAS, PA Dept. of Agriculture, National Dairy Council, PT Indesso Aroma, and Sherwin-Williams Company. HH has received travel support and honoraria from AOAC, University of Alabama, and Henkel Adhesive Technology for speaking at meetings. HH is the Rasmussen Career Development Professor in Food Science, a named professorship made possible by a philanthropic gift by Frederick, Sr. & Faith E. Rasmussen. ALB is or has been a member of the following professional societies: IFT, SSP, Gamma Sigma Delta, Phi Tau Sigma, and the Philanthropic Educational Organization International. AJB is or has been a member of the following professional societies: IFT, SSP. HH is or has been a member of the following professional societies: ASEV, IFT, FCIA, SSP, GöCH, ASAC, E3S, and Gamma Sigma Delta. None of these organizations nor the funders have had any role in the design of the study; in the

between 1 metric ton and 200 metric tons of cocoa beans per year and is at least 75% owned by the company itself or the company's employees)" [1 p. 31]. In the intervening time, several companies took on investors for growth purposes and by 2017 a more accurate definition was provided by Leissle [4], which we use for this paper [2]. A craft chocolate company (a) is one that starts with cocoa beans and produces finished chocolate ("bean-to-bar"); (b) is not owned by one of the "Big Five" multinationals (Mondelēz International, Inc; Ferrero-Rocher SpA; Nestlé SA; The Hershey Company; Mars, Inc.) [7]; (c) and was established during the recent wave of innovation, since 1996 [4 p. 39].

## Chocolate market

The United States chocolate retail market is valued at more than $19 billion by market research firm Mintel [8]. The National Confectioners Association (NCA) serves as the main lobbying group for the confectionery industry and tracks chocolate purchases using retail measurement and household data as a service to its constituents. In their product tracking, the NCA traditionally divided chocolate products into everyday chocolate, priced at less than $8 per pound, and premium chocolate, priced at more than $8 per pound. In 2016, the NCA increased the price demarcating premium chocolate to $11 per pound [9, 10]. Then, for their 2019 survey of chocolate consumers, the NCA changed the category name from everyday to "mainstream" and added a third category, "fine chocolate" [9]. According to the NCA, mainstream chocolate is typified by Hershey, Snickers or Baby Ruth; premium chocolate is typified as Lindt, Ghirardelli or Ferrero; and fine chocolate is that "made by small artisan chocolatiers, who source the best quality cacao, create small-batch products with unique flavors and textures, and educate consumers about the product and process" [9]. Ultimately, fine chocolate companies as defined by the FCIA include not only the makers of what we have defined as craft chocolate, but also chocolatiers, companies that produce chocolate used by chocolatiers, and multinationals that own craft chocolate brands [11].

In 2016, Vreeland & Associates valued the craft chocolate industry at $100 million, which is no longer accurate because of the industry's rapid growth [13; C. Vreeland, personal communication, September 20, 2018]. It is difficult to put a value on the fine chocolate and craft chocolate market because few of the purchases are captured by retail measurement and household data, which rely on scanning Universal Product Codes [9]. Regardless, craft chocolate comprises a substantial portion of the US chocolate market [12]. In addition, craft chocolate has already been the target of multinational confectionery company acquisitions, and it is reasonable to predict that craft chocolate will follow the acquisition trend of craft beer [3, 13–15].

Because of craft chocolate's value to the confectionery industry, its attractiveness to multinationals, and its alignment with other craft industries (namely, craft beer, artisanal cheese, and specialty coffee), it presents an interesting case study to understand this new movement of craft consumption. Additionally, there is no previous literature on motivators for craft chocolate perception and purchase. Because premium chocolate consumers are the most likely segment to trade up to craft chocolate, this study sought to uncover premium chocolate consumer perceptions of craft chocolate and desirable chocolate attributes [16].

## Research objectives

We had the following research objectives:

1. To gain insight into American premium chocolate consumer perception of craft chocolate.

2. To identify search, experience, and credence attributes that are important to American premium chocolate consumers.

collection, analyses, or interpretation of data; in the writing of the manuscript, or in the decision to publish the results. The competing interests statement in the manuscript does not alter our adherence to PLOS ONE policies on sharing data and materials.

## International chocolate consumers

Most chocolate consumer behavior studies have been carried out in Europe to identify consumer perception and willingness to pay for sustainability labels (e.g. organic, fair trade). Studies in Germany, the UK, France, Italy, and Belgium showed that consumers segment based upon demographics or psychographics in terms of their interest in purchasing sustainably-labeled chocolate [17–24]. UK chocolate consumer research concluded that the flavor and brand of the chocolate bar must be congruent to elicit a positive emotional response and recurrent purchases from consumers [21]. A recent study with Australian chocolate consumers found that packaging was a strong driver of consumer expectations and liking [25]. Overall, these studies determined that consumer attitude and behavior towards chocolate product attributes are strongly linked to consumer location and cannot be transferred to consumers from other countries.

## American chocolate consumers

American chocolate consumers are regularly surveyed by Mintel in their biennial Chocolate Confectionery, US Reports [16, 26]. These reports provide a barometer of year-to-year shifts in chocolate purchase behaviors, but do not pursue the "why" behind consumer choices.

One of the only academic studies on American consumer perception of chocolate compared younger and older Midwestern millennial preference for sustainability certifications on candy bars using focus groups followed by a choice experiment [27]. They determined that younger millennials (ages 18–25 years old) were primarily focused on taste in their candy bar purchases and were not interested in sustainability certifications, while older millennial (ages 26–35 years old) cited positive views towards sustainability certifications in focus groups, but these attitudes did not align with the results of the choice experiment [27]. One of the main reasons consumers cited as a motivator for purchasing chocolate with sustainability certifications was guilt reduction by making healthy choices for themselves (organic) and also better working conditions for cocoa producers (fair trade) [27]. While this study scratched the surface of American consumer attitudes towards sustainability certifications in chocolate, it investigated these in the context of candy bars. In addition, the choice experiment presented 435 pairings to each participant and was reported to have exhausted participants, which makes it difficult to interpret the findings.

## American craft chocolate consumers

Academic literature specific to craft chocolate has investigated the use of the word "artisan" as a brand and the rejection of sustainability certifications by craft chocolate makers [4, 5]. In her survey of 100 attendees of the 2014 Northwest Chocolate Festival, Leissle [4] discovered that 48% of "interested" chocolate consumers defined the difference between "artisan" and "industrial" chocolate as the "flavor of the chocolate bar." Leissle [4] concluded that consumers buy "artisan" chocolate to resolve their moral conflict between enjoying a middle-class luxury, like chocolate, and buying from an exploitative value chain [4]. Woolley et al. [5] found that the majority of craft chocolate companies reject the use of sustainability certifications because they believe that direct trade is preferable and a sustainability certification would dilute their brand. In direct trade, cacao sourcers go to the producing country, discuss post-harvest processing with the farmer, and typically pay higher prices for their cocoa beans than the fair trade standard [5].

As a service to their industry members, chocolate industry associations have characterized craft chocolate consumers. In 2017, the FCIA administered 1,000 surveys at chocolate shows across the US and conducted focus groups in various chocolate shops with over 120 consumers

[6]. Their sample population was not the general public, but "chocolate enthusiasts" and "connoisseurs" who demonstrated a strong interest in fine chocolate. As a continuation of this work, FCIA collaborated with the NCA in 2019 to broaden their scope with a national survey of 1,500 chocolate consumers of all types in an effort to understand differences among consumer segments [9].

NCA survey results showed that 27% of chocolate consumers identify as fine chocolate consumers, and of this population, sustainability certifications are most desired by millennials for whom cacao farming and chocolate production practices are important [9]. This finding opposes Young & McCoy's [27] overall finding that Midwestern millennial consumers (who were not screened based upon level of engagement) are not interested in sustainability labels. The FCIA study with chocolate enthusiasts found that fair trade was preferred more than direct trade [6].

The FCIA study found that the top purchase motivators for fine chocolate were pleasure, gifts, health, and environmental impact, while the NCA study reported that among fine chocolate buyers, the top motivations for purchasing fine chocolate were that it tastes better, is more satisfying, makes a good gift, and supports small businesses [6, 9]. The most influential factors for chocolate purchases among all chocolate consumers surveyed by the NCA were mood, brand, and price [9].

In characterizing fine and craft chocolate consumers, the NCA found that fine chocolate consumers are younger, more affluent, likely to live in urban areas, and greatly value social and environmental stewardship [9]. Interestingly, core fine chocolate consumers (defined as the 11% of the survey population who regularly purchase fine chocolate) were more likely to believe American-made chocolate is better that European chocolate, whereas premium chocolate consumers were more likely to believe that European chocolate is better [9]. Chocolate enthusiasts overwhelmingly preferred dark chocolate and cacao percentage was important to 73% of fine chocolate consumers [6, 9]. Experimentation and trying novel chocolates was found to be essential to fine chocolate consumption [9].

While the findings of these two surveys and focus groups help to understand the views of craft and fine chocolate consumers, the implications are limited. The FCIA work was restricted to consumers already purchasing craft chocolate, while the NCA study segmented between fine chocolate and premium consumers, but their definition for fine chocolate was vague and not synonymous with craft chocolate. Both surveys fall victim to typical limitations of survey data, which is that they do not probe deeply to understand consumer emotions and feelings behind purchases. In addition, the focus groups were not analyzed using a robust method, such as coding, and instead were only summarized.

## Methods

The research met the criteria for exempt research according to the policies of this institution and the provisions of applicable U.S. Federal Regulations. This study was reviewed by The Pennsylvania State University Institutional Review Board and was deemed exempt by exemption category six (taste and food quality evaluation; protocol number 6654). All participants provided informed, oral consent and were compensated for their time ($10/hour).

### Study design

To understand desirable chocolate attributes and how premium chocolate consumers perceive craft chocolate, we developed a mixed-methods study that used focus groups and a projective mapping activity.

**Rationale for focus groups.**   Because no previous literature or reports describe American premium chocolate consumer attitudes to craft chocolate or desirable chocolate attributes, an

exploratory method such as a focus group is an appropriate method to fulfill our research objectives and generate hypotheses for future work [28]. In addition, a focus group allows us to look for a range of ideas or feelings that premium chocolate consumers have about chocolate and uncover factors that influence opinions, behaviors, and motivation. Additionally, a focus group familiarizes us with consumer language related to premium and craft chocolate [28].

**Rationale for projective mapping.**   In conjunction with the focus group, a projective mapping activity was carried out in advance by focus group participants and used as a visual aid during focus group introductions [29]. Projective mapping is a rapid technique in which consumers place products on a blank space in terms of their relationship to one another (e.g. flavor, quality, etc.) [29]. In the field of food and consumer science, qualitative methods, such as focus groups, are sometimes regarded as less robust than quantitative methods because participants may be influenced by social bias and not express their honest personal opinions [30–32]. One solution, as Risvik et al. [29] suggested, is the use of a focus group combined with a projective mapping activity, wherein the quantitative mapping activity may be discussed during the focus group and used to validate the focus group findings. With this work, we sought to demonstrate that chocolate products could be mapped by consumers, that the map could be used during the focus group itself as an introductory visual aid, and that the map could be analyzed later and used as a tool to compare with and enhance focus group findings.

**Participant selection.**   Participants were recruited via email using two electronic mailing lists with voluntary subscribers, consisting of employees, students and community members of The Pennsylvania State University campus and surrounding area (State College, PA). Potential participants completed an online screener for eligibility and willingness to participate, created in Compusense Cloud software (Compusense Cloud, Academic Consortium, Guelph, ONT, Canada). In an effort to find engaged premium chocolate consumers, screening criteria were developed using best practices as described by Stone, Bleibaum, & Sidel [33], and to be more stringent than the Mintel chocolate consumer criteria of "18+ and purchased chocolate within the last three months" [16]. The criteria used in this experiment were as follows: between 18–70 years old; not pregnant or breastfeeding; no food allergies or sensitivities to chocolate; fluency in English; primary food shopper; frequent chocolate consumption (from daily to two to three times per month); weekly to monthly consumption of premium chocolate (Godiva, Lindt, Guittard, Eclat, Dandelion, Ghirardelli, Vosges, etc.); and articulateness as determined by answers to an open-ended question about the participant's "most memorable chocolate moment." Of the 625 subscribers who filled out the screener, a total of 27 (15 females) reportedly healthy individuals, ages 22–67, were selected as participants, provided informed consent, opted to participate, and were compensated for their time ($10/hour). All participants who were selected chose to participate.

**Projective mapping design.**   One week prior to the focus group, participants were assigned a projective mapping activity to complete individually at home [29, 34–36]. Each participant received a 17-inch (43.2 cm) by 11-inch (27.9 cm) sheet of plain white paper and 47 stickers (see see Image 1 https://doi.org/10.26207/a863-pp02) of various chocolate products that ranged from mainstream to premium and craft chocolates. As with other projective mapping studies with non-taste stimuli, participants did not eat the products [37]. Participants were instructed to do the following:

> *Please evaluate the chocolate products (on the stickers) according to similarities or dissimilarities in quality attributes by placing similar samples close to each other and more dissimilar samples further apart on the attached large sheet of paper. Once you reach a final configuration, note down appropriate descriptors for the characteristics of the chocolates, directly on the*

*large sheet of paper, when needed. In addition, if the final configuration contains groupings, please label and/or circle the groupings as you see fit.*

The term "quality" was used in an effort to provide clear, yet non-directing instruction in order to promote nuanced and individual groupings reflective of participant's preconception of desirable attributes. Participants brought the sheet of paper to the focus groups and used it as a visual aid for "show and tell" to describe how they perceive chocolate quality.

**Focus group design.** A total of four focus groups lasting approximately 120 minutes each were conducted with 5–8 participants in January and February 2017. Each focus group was led by the first author and observed by the second and last authors. Discussions took place in The Pennsylvania State University Department of Food Science Focus Group Room, a custom-built qualitative research facility located in the Erickson Food Science Building (University Park, PA, USA). All discussions were audio recorded with voice recorders (Sony ICDPX370, New York, NY, USA). The audio recordings were transcribed verbatim by a commercial transcription service (Landmark Associates, Phoenix, AZ, USA) and checked against the audio recordings to verify completeness. Immediately following each focus group, the moderator and two observers mapped each focus group for themes.

Because chocolate has been considered an aphrodisiac and American women have reported craving chocolate during perimenstruation, focus groups were divided by gender into two groups of all men and two groups of all women, to follow best practices and allow participants to feel comfortable discussing chocolate [38–42]. Aside from basic demographic data collected from the participants based upon information they provided to the database, no information on profession, education, income, or otherwise, was collected from the participants.

The focus group questioning route was divided into three parts and sought to encourage a comfortable discussion of the research questions by beginning with general questions that eventually narrowed in focus (See S1 Appendix for Focus Group Moderator Guide) [38, 43]. In part one, participants introduced themselves, recalled their most significant chocolate moment that they had articulated in the screener, explained their projective maps to the group, and described how they interpret chocolate quality. In part two, participants tasted five different chocolate bars, one at a time, that presented a range of commercial chocolates (see Image 2 https://doi.org/10.26207/a863-pp02) [9]. Due to the ubiquitous presence of the Hershey's milk chocolate bar (The Hershey Company, Hershey, PA, USA) in the United States, it was selected to represent mainstream chocolate [9]. Lindt 70% cocoa dark chocolate (Lindt & Sprüngli USA, Inc., Stratham, NH, USA) was selected to represent premium chocolate and a Swiss and European bar. Green & Black's Organic Dark Chocolate Bar 70% Cacao (Mondelēz International, Inc, East Hanover, NJ, USA) was selected because it is also premium and has a United States Department of Agriculture (USDA) Organic and a Fair Trade label. Endangered Species Dark Chocolate with Sea Salt and Almonds (Endangered Species Chocolate, Indianapolis, IN, USA) was selected as a third premium chocolate because the chocolate bar supports a cause, with 10% of proceeds benefitting an endangered species on the packaging [44, 45]. In addition, Endangered Species Chocolate prominently displays Non-GMO Project, Fair Trade, Gluten Free, and Vegan certifications. Endangered Species Chocolate has since changed its label due to internal consumer and ethnographic research [44, 45]. Dandelion Chocolate 70% Ambanja, Madagascar (Dandelion Chocolate, San Francisco, CA, USA) was selected to represent craft chocolate because Dandelion is one of the original craft chocolate companies and a segment leader [2]. We considered using a counterbalanced order for sample presentation, but decided against it for both simple logistical reasons (i.e., any blocking would need to occur at the order of group, not individual, to avoid mixing up samples), and more critically, because we were concerned about contrast effects and carryover effects that would occur if a group

tried a simple mild chocolate immediately after an intense and complex chocolate, as carryover and contrast would obscure the flavor of the simpler milder chocolate [46, 47]. While not yet established in the sensory literature for chocolate, the same logic of least intense to most intense is used in wine tastings because it has been shown to dramatically impact perception of wine flavor [48]. All of the chocolate bars presented were also used as stickers in the projective mapping activity. Due to availability, a different origin was selected for the Dandelion Chocolate bar, but the package looked almost the same as the sticker used (see Images 1 and 2 https://doi.org/10.26207/a863-pp02). Participants sampled one piece of each chocolate bar at a time and were given time to write down notes. Afterwards, in a group discussion, they described the product in terms of flavor, packaging, certification labels and other elements and attributes they found appealing and unappealing. For part three, participants discussed what encourages them to purchase a new food product and shared words they would use to describe the chocolates tasted. With four focus groups in total it was possible to identify key themes for the project within financial constraints [49–51].

## Data analysis

**Focus group word clouds.**   For part two of the focus groups, each transcript was segmented by chocolate product discussed (i.e. Hershey's, Lindt, Green & Black's, etc.) and combined across all four focus groups. These text segments were analyzed using Voyant Tools (version 2.4) to remove stopwords, create frequency tables, and ultimately, word clouds [52]. Stopwords were defined as function words that do not carry meaning (e.g. won't) or words that were used at such a high frequency that their meaning did not differ across corpora (e.g. chocolate) [52]. Texts were cleaned using the Voyant Tools standard English stopword list of 485 words, which was edited to include additional words from the corpora. Word clouds were constructed from the top 95 words used for each chocolate bar.

**Focus group data analysis.**   Focus group transcripts were analyzed using grounded theory and inductive methods with an emphasis on emergent themes [50, 53]. Grounded theory is an appropriate method because craft chocolate is a novel food product and there is no prior research from which to preconceive codes [50, 54].

Coding was performed by two researchers who read through one transcript and agreed on a master codebook relevant to the research questions [55]. The "classic approach" otherwise known as the "scissor-and-sort" technique, was used to cut up the printed transcripts, group similar quotes, and then assign the quotes to codes [50, 55–57]. Particular attention was given to quotes where participants showed emotion, enthusiasm, passion or intensity [55, 57]. While topic frequency in quotes was observed, it was not a mandatory criterion for coding and outlier quotes were acknowledged [57]. Once codes were determined, they were assembled into memos and memos were subsumed into themes [50]. Themes were back-checked for consistency, coherency, and distinctiveness among researchers and with the thematic maps that were produced at the end of each focus group.

**Projective map data analysis.**   Projective maps were analyzed using Multiple Factor Analysis to create a product map with a product descriptor overlay [58]. Three participants did not follow the instructions correctly and their product maps were unusable, leaving 24 product maps to be analyzed. First, x, y-coordinates of the stickers were measured using the bottom left corner of the map as the origin [59, 60]. Words that were used to describe the products in the maps were coded by three researchers who agreed upon a master codebook. The three researchers coded 12 of the 24 total maps together and the remaining maps were coded by one researcher. One sample, the Raspberry Ghirardelli tablet, was dropped from the analysis due to a coding error. The final chocolate product number was 46.

Data was analyzed in R (version 3.5.1) [61] with RStudio (version 1.1.456, Boston, MA, USA) using the FactoMineR and SensoMineR packages [62, 63]. A contingency table was created using X, Y-coordinate data as the first 46 columns and the code words as the last 24 columns per product. Multiple Factor Analysis [58, 63, 64] was used to interpret the product space with the qualitative data by running the code words as supplementary variables [58, 63, 64]. Word Count Analysis [62] was used to identify consensual words, which are defined as words that have the same meaning for most of the participants at a significant level ($p < 0.05$) of consensus [65].

## Results & discussion

The following section outlines focus group and projective mapping analyses as answers to the research objectives. Each results section contains a brief discussion section.

### American premium chocolate consumer perception of craft chocolate

Overall, premium chocolate consumers perceived craft chocolate as novel and exciting, and struggled to contextualize it. Consumers were surprised by the fruity flavor and likened craft chocolate to coffee and wine in terms of flavor and packaging elements. In comparison to the mainstream and premium chocolate bars, when the Dandelion bar was revealed, participants were anxious to try it, and in focus group four, participants squealed with delight. For most focus group participants, this excitement translated into a quality determinant, which was expressed by a male consumer:

*Wow, that must be good. I don't even recognize it.*

The word clouds in Fig 1 illustrate different words used to describe each chocolate bar among the four focus groups. For the Hershey bar (Fig 1A), the focus was on its "sweet" "taste," "smooth" "melt," and "creamy" texture. The second bar tasted was the Lindt 70% bar (Fig 1B), for which the words "dark," "taste," and "bitterness" are noteworthy. Additionally, when this bar was discussed, "packaging" was important and participants across all focus groups focused on the predominance of "70" "percent" "cocoa" as an indicator of quality. For all four focus groups, the conversation around the Green & Black's bar (Fig 1C) was about "organic" and "fair" "trade." With the Endangered Species chocolate bar (Fig 1D), the meaning of "GMO" was questioned in every focus group. Participants also focused on the inclusion of "sea" "salt" and "almonds" in the chocolate bar. For the Dandelion chocolate bar (Fig 1E), participants described the bar as "different" and wondered where the "beans" came from. The discussion of "flavor" was accentuated by use of the words "fruity," "wine," and "coffee." Additionally, the name "greg," the cocoa sourcer listed on the back of the package, was prominently discussed. The meanings of these words will be discussed in the following sections.

A few words came up in the discussion of every chocolate bar. The word "mean" appears in every word cloud and dominates some of them. Participants questioned the meaning of several packaging elements (i.e. What does it mean?), such as sustainability labels (non-GMO, organic, fair trade) and cocoa percentage. Additionally, "Hershey" appears in every word cloud. Participants compared all of the chocolate bars to the Hershey bar in terms of taste, flavor, safety, sustainability, and claims. This may have been because the Hershey bar was tasted first in the sequence or because the Hershey bar was the most familiar chocolate bar to participants.

The difference in word use frequency during discussions of the different chocolate bars illustrate that when tasting the mainstream, and only milk chocolate bar in the group, the

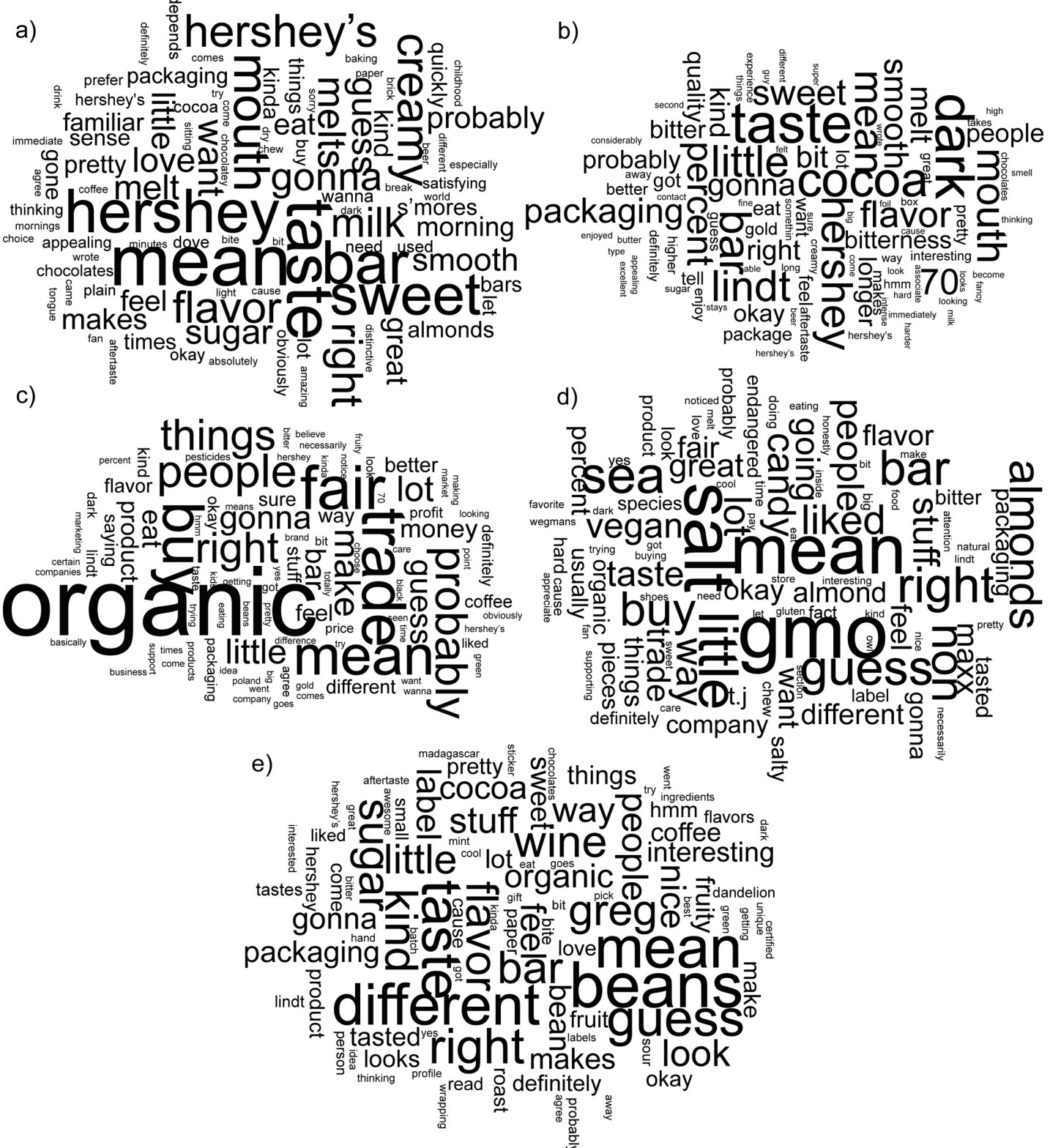

**Fig 1. Word clouds created from the 95 most frequently used words for the sampled chocolate products in the focus groups.** a) Hershey's milk chocolate, b) Lindt 70% cocoa dark chocolate bar, c) Green & Black's Organic Dark Chocolate bar 70%, d) Endangered Species Dark Chocolate with Sea Salt and Almonds, e) Dandelion Chocolate 70% Ambanja, Madagascar.

emphasis was on the sweet taste and creamy, smooth melt of the Hershey bar, whereas with the Lindt, Green & Black's and Endangered Species bars, the focus was on packaging elements, such as cacao percentage, organic and GMO [free] certifications. When the Dandelion craft chocolate bar was tasted, participants focused on flavor. Several participants were curious how the fruit flavor got into the chocolate and could not believe that the ingredients statement did not include raspberry.

Participants struggled to contextualize the Dandelion craft chocolate and likened it to non-chocolate products, such as cologne, bath products, wine or coffee. Several consumers noted the uniqueness of the package. They decided very quickly that there was a personal affect to the product where it stated that the roast profile was created by Chiann and the beans were sourced by Greg (see Image 2 https://doi.org/10.26207/a863-pp02). A female consumer said:

> *The thing it reminds me of a lot though is like Lush cosmetics and soaps and stuff. Since on their products the person who makes the actual soap has a sticker with their face on it in a cartoon. They stick it on the side and it's like, "Made by Gerry" or this one was made by Susan on this day at this time. Something about that I always think is really entertaining. It's kind of the same thing to me. You get an idea that there was a person who actually made this.*

Consumers compared craft chocolate to wine or coffee, conceptually, in terms of flavor, packaging, sustainability labels, and mouthfeel. They interpreted the cacao origin as the vineyard and the cacao variety as the grape variety. A male consumer said of the Lindt chocolate bar:

> *[It] was more adult flavor, much better by any measurable standard, I think. Kinda sweet for me. It's more like comparing chess to checkers, maybe. You know what I mean? The flavor profile's more multidimensional—it was more like when you're drinkin' wine or somethin' like that.*

We expected consumers to be excited by craft chocolate because the NCA survey found that novelty is one of the main drivers for fine chocolate purchases [9]. We were surprised by the comparisons consumers made with cologne, bath products, wine, and coffee. It is noteworthy that these comparison product categories can also be divided into mainstream, premium and craft segments.

## Important attributes for American premium chocolate consumers

Overall, focus group participants used a variety of attributes to judge the quality of chocolate bar stickers that were mapped and products consumed during the focus group. Surprisingly, even when participants ate the chocolate bars, they used mostly "extrinsic cues," such as packaging, rather than "intrinsic cues," such as flavor, to judge product quality [66]. This is in line with much of the wine marketing research [67–69], but opposes classic consumer behavior works, which cite the importance of intrinsic cues in products such as meat, and consumer goods like ground coffee and shampoo [66, 70]. The attributes found desirable by American premium chocolate consumers can be organized using the framework provided by Darby & Karni [71] who renamed extrinsic attributes as "search" and intrinsic attributes as "experience" and added a third "credence" attribute, for which the true value cannot be verified and they are impressed upon the product by the consumer [71, 72]. Thomson et al.'s [20] conceptualization framework compliments that of Darby & Karni [71] wherein abstract conceptualizations are credence attributes, which in our study is trust; functional or experience attributes which in our study is utility; and emotional, also an experience attribute, which in our study is joy.

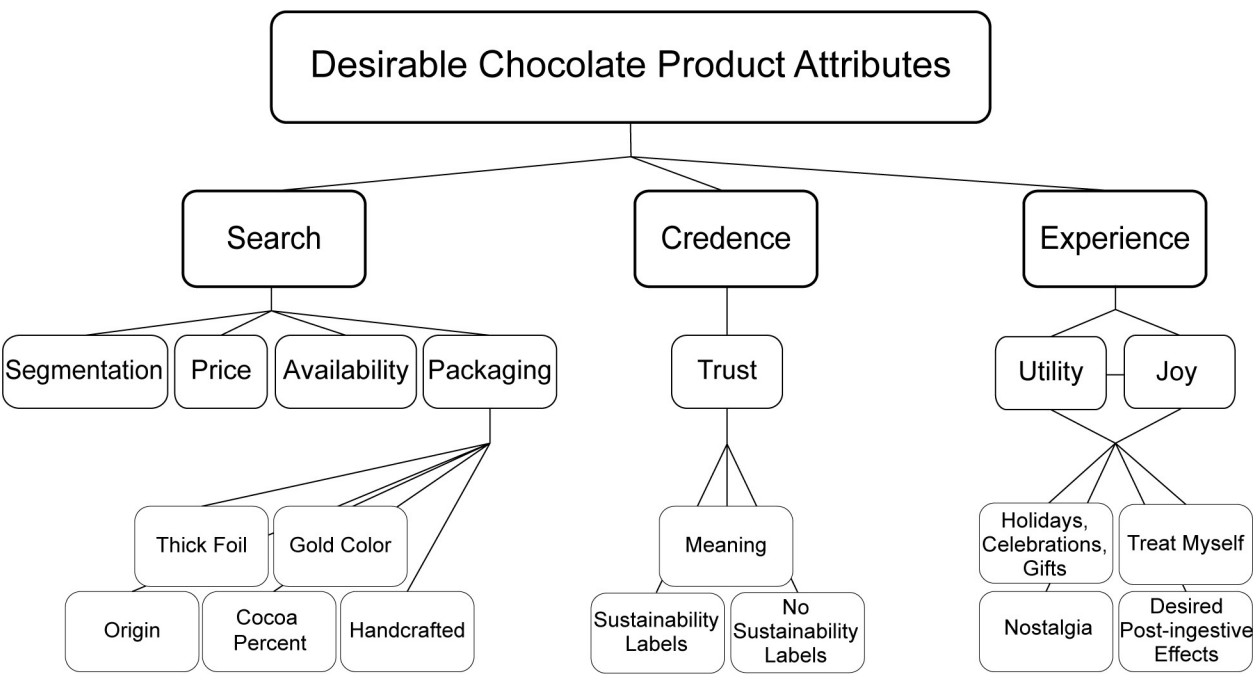

**Fig 2. Desirable chocolate product attributes for premium chocolate consumers.**

The relationship among these attributes are shown in Fig 2 and will be discussed in the order of search (i.e. segmentation, price, availability, packaging), credence (i.e. trust), and experience (i.e. joy and/or utility) attributes below.

### Search attribute: Segmentation

Fig 3 shows the product space produced by the Multiple Factor Analysis of the 24 usable participant projective maps. In the map, there are three main sections that are aptly described by the consensual words shown in bolded red: cheap, American, available, and candy on the bottom right quadrant; specialty, artisan, fair trade, organic, flavor, and dark chocolate on the bottom left; and then individually-wrapped and special occasion on the top middle. These segments will henceforth be called candy, premium and special occasion chocolate.

Focus group discussions further elucidated how participants segmented the chocolate products. Instead of using marketing jargon, participants separated the chocolate products based upon age-appropriateness, calling the bottom right or candy segment, "kids" chocolate; the bottom left or premium chocolate, "adult" chocolate; and the top middle segment, special occasion chocolate, "grandma" chocolate. A female consumer described how she differentiated the bottom right and bottom left quadrants:

*Hershey goes for kids, bright, I think of them as playground colors where these are more coffee shop setting. Like a little more fancy.*

A male consumer explained his map, saying:

*Over here is Grandma's chocolates, the Whitman's sampler. You go to old folks; this is what they buy.*

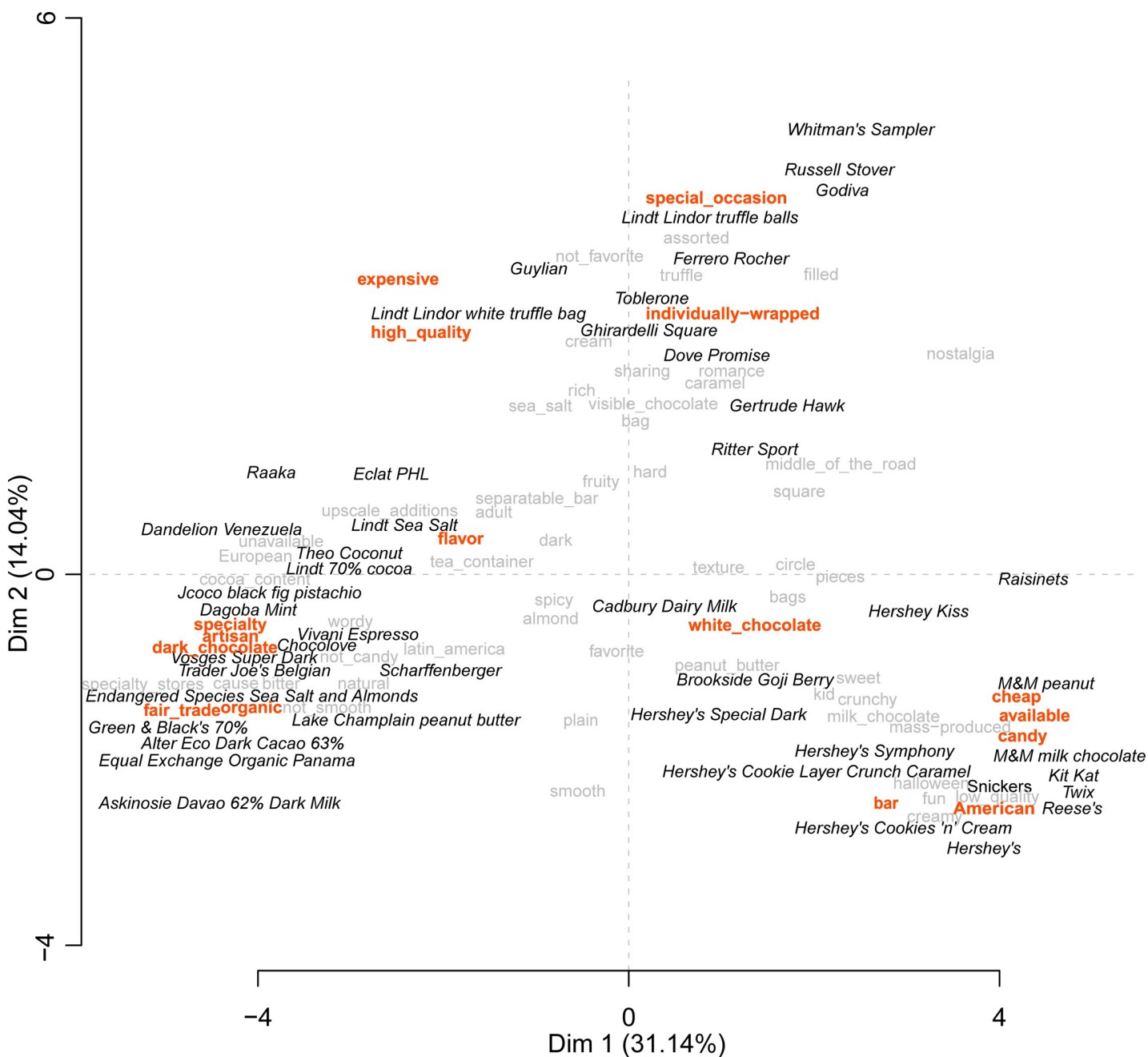

**Fig 3. Product map created from projective mapping task.** Chocolate products are shown in italicized black, consensual words ($p < 0.05$) are in bold red, and non-consensual words are in gray.

The candy segment is characterized by the presence of Hershey and Mars products, such as the Hershey Kiss, M&M products, and classic Hershey bars. The premium segment is composed of what the NCA considers premium chocolates and fine chocolates [9]. An interesting finding is that the segments that consumers created, candy, premium, and special occasion, differ from the NCA segments of mainstream, premium and fine chocolate, because they were organized by purchase and/or eating occasion rather than price. Participants placed special occasion chocolate in its own segment, which the NCA bundles together with premium

chocolate [9]. Additionally, craft chocolates mapped directly onto premium chocolates and were not differentiated by premium chocolate consumers, indicating that premium chocolate consumers place craft chocolate in the same category as premium chocolate.

## Search attribute: Price

Price was a large factor in separating the products. On the top left quadrant of the product map, the words "high quality" and "expensive" overlay one another and are located opposite the map from the word "cheap." This indicates that price and quality are highly associated: if the chocolate is expensive, it must be high quality; if the chocolate is cheap, it must be low quality. A female consumer explained:

> *These are a little more sophisticated and so my brain just thinks they must taste better they must be higher quality. Also, they're more expensive so I think it's better.*

This finding aligns with previous research, which shows that across several products, including wine, price is considered one of the most important extrinsic product cues [67, 73–78].

The phrases "high-quality" and "expensive" are equidistant from premium and special occasion chocolate, which indicates that both of these segments of chocolate are perceived as expensive and high quality. The eloquent relationship between expensive and high-quality demonstrates Lichtenstein et al.'s price-quality schema, or the consumers' propensity to use price to make generalized attributions about the product [74]. The close distance of "expensive" and "high quality" to the special occasion segment, which is usually gifted, likely indicates prestige sensitivity, where a consumer purchases an expensive and high-quality chocolate to give to someone as a gift and demonstrate their expensive taste [74].

Our findings also align with those of the NCA survey, in which American non-core fine chocolate consumers listed "expensive" as their one-word first impression of fine chocolate [9]. This was also true of young Finnish and Australian chocolate consumers who believed that price was a good indicator of chocolate quality [79, 80].

## Search attribute: Availability

In general, the location where chocolates can be purchased was a quality indicator for participants. On the product map, "candy" is located next to the word "available", which was a code word for locations such as the gas station, drug store, supermarket, and movie theater, where chocolate products are readily-available. The code word "unavailable", while not consensual, is located among the premium chocolates. If the chocolate is readily-available, then it is low quality; if the chocolate is obscure, difficult to find, or only found in a few shops or online, it is high quality. A female consumer explained:

> *. . .Then I had this weird section that was like middle of the road. You could definitely find it at a [national drug store]. You might not find it at a gas station, but still pretty accessible and affordable but higher quality than a Hershey's bar. Then I had the more exclusive, like this might be harder to find. Where you might have to go to a [regional, traditional supermarket] or a [national natural market] or someplace even just a specialty store. I feel like some random—you can find these at [a national discount store] where they have the nice chocolate for some reason.*

A male consumer clarified the connection between ready availability and price:

*[T]he more traditional chocolates, because they're so readily available. They're not hard to find. They might even be less expensive.*

Product scarcity as an indicator of quality is a well-known concept in consumer literature [81]. Dandelion Chocolate has admitted that they do not sell their chocolates at drug stores in San Francisco because it would directly conflict with Dandelion's aspirations to be seen as an "exquisite" and "craftsman" brand [2]. Instead, Dandelion Chocolate is available online, in the Dandelion cafe, and from specialty shops, which maintains their exclusivity [2]. Research with Mexican craft beer consumers found that some consumers would like to find craft beer everywhere, whereas for others, the act of going out and looking for beer in small, specialized stores is part of the craft experience [82]. In the NCA survey, chocolate consumers selected all of the locations where they purchase chocolate [9]. Nearly half of mainstream chocolate consumers purchase most of their chocolate at grocery stores compared to only a quarter of premium chocolate consumers who are also far more likely to purchase their chocolate at specialty chocolate shops [9].

## Search attribute: Packaging

The importance of packaging was demonstrated in the projective mapping activity, where it was the only attribute available to judge quality on the stickers, and in the focus group itself when consumers had a chance to look at five of the chocolate packages up close. In these two activities, it became clear that product packaging is critical to first impressions, initial and ongoing product interactions, and the formation of long-lasting relationships between the brand, product, and consumer [83]. Participants were clear that if they have not tasted a chocolate before, then they rely on packaging to convey whether that product is a worthwhile purchase. A male consumer stated:

*A big one for me is just packaging, too, if you can't try it before you buy it. I do look at the packaging, and I will choose something just solely based on what it says on it, obviously.*

Our study findings are consistent with Australian chocolate consumer attitudes that product liking is affected by the expectations generated by the packaging [25]. Throughout all four focus groups, participants opted to discuss packaging extensively and appeared to have a higher emotional attachment to the packaging than the taste of the product [25].

**Chocolate or cocoa origin.** Participants considered European chocolates to be higher quality than American. In the product map (Fig 2), the descriptor "European", while not consensual, is geographically closer to the word "quality" than the descriptor, "American." This reinforces the focus group discussion in which a female consumer described her product map:

*I said basic American candy bars, but I like Ghirardelli a lot also and Lindor. I like Lindt and Ritter. I like Ritter bars. Yeah, they're not as sweet, I don't think. It's more like the European where you get the rich chocolate and you don't get just that sugary—where, I think, the American bars, you really just—sometimes it's more of a sugar high than a chocolate high.*

The NCA survey found that 45% of premium chocolate consumers believe European chocolate is better than American, versus 39% who thought there was no difference and 16% who believed American-made is better [9]. A Belgian choice experiment revealed that country of manufacture was very important to chocolate lovers and was one of their most important considerations when purchasing chocolate [18]. Belgian chocolate was highly-preferred in the study, likely due to nationalistic tendencies and Belgium's history of having invented the filled chocolate bonbon or praline [18, 84]. While many studies have investigated consumer

preference for cocoa country of origin, this was not addressed by consumers in our study [24, 85]. Premium chocolate consumers may be more familiar with traditional countries of chocolate manufacture, such as Belgium and Switzerland, rather than the flavor associated with origin chocolates. Leissle explains that "Belgian chocolate" has an advantage in the marketplace over "Ghanaian cocoa" because chocolate eaters have become accustomed to flavors associated with characteristic chocolate styles rather than flavor associated with cocoa origin: Swiss is creamy from additional cocoa butter, Belgian is soft milk, while American is milky or slightly sour [86].

**Handcrafted.** Consumers distinguished between products that appeared handcrafted versus commercial chocolates. The distinction was made based upon the packaging, type of chocolate (dark or milk), and flavors. A male consumer said:

*I think it's also to give you a feel of these are made in small batches by hand, and these are packaged by hand, and these labels are applied by hand.*

Another male consumer explained the commercial chocolates on his projective map:

*There's a sort of a corner for mass market chocolates that had generally a lot of milk, sugar, or other additives, not necessarily that much cocoa.*

This finding aligns with Leissle's [4] work on the use of "artisan" as brand by craft chocolate makers. Her textual discourse of 129 craft chocolate websites found that 49% of craft makers self-name as "artisan" and the only common definition for artisan is that it is not "industrial" [4]. If craft chocolate makers succeed at communicating a hand-crafted aesthetic, they effectively convey that they are small in size and not industrial.

**Cocoa percentage indicates high quality.** Consumers stated that products with the cocoa percentage written on the package indicates that the chocolate bar is high quality. While not found to be a consensual term, "cocoa content" was mapped among the premium chocolates in the product map. A male consumer said:

*Just from a packaging standpoint, I like that they say, "70 percent." I think that speaks to the quality of the bar. I thought it was a rich chocolate bar. You tasted the cocoa in the bar, which I think speaks to the quality of the chocolate, too. It was sweet, but there was that typical dark chocolate bitterness with it, which I love dark chocolate, too. I thought it was great.*

In the NCA survey, 53% of premium chocolate consumers and 73% of fine chocolate consumers said that cacao percentage was important and had a significant influence on their purchase decisions [9]. Premium chocolate consumers preferred a cacao percentage between 71–80%, whereas fine chocolate consumers preferred percentage levels of 75% or more [9]. In the FCIA consumer work, chocolate enthusiasts overwhelmingly preferred dark chocolate and cited cacao percentage as very important to their purchase decision [6]. In a Spanish consumer test that compared the same chocolates with and without packaging, three consumer segments emerged: consumers who would buy a product because it has cacao percentage listed, consumers who are indifferent to cacao percentage, and consumers who are less likely to buy the chocolate because the cacao percentage is listed [87].

Chocolate with cocoa percentage written on the label is typically dark and consumers strongly associated dark chocolate with quality. A female consumer explained:

*I think of milk chocolate as lower quality. I don't know why but dark chocolate for me is more exquisite. . .*

Chocolate consumer focus groups held in Pennsylvania and New York also found that participants associated dark chocolate with good taste and social distinction [88].

**Gold color indicates high quality.**   Participants focused on the color gold incorporated into packaging as an indicator of high quality and high price. The lettering on the Lindt 70%, Green & Black's, and the Dandelion package is gold, and the inner foil wrap of the Green & Black's bar and the Dandelion bar is gold (see Image 2 https://doi.org/10.26207/a863-pp02). While the main colors of the Lindt package are black and white, and the Green & Black's package is black and brown, gold is the dominant color of the outer and inner packaging of the Dandelion bar. A female consumer said of the Lindt 70% bar:

*Yeah. The packaging, the gold on there I mean it's a little fancy. You need to like this because it is in gold. [Laughter] You know? This is high class, really good.*

Several consumers referenced the fictional character Willy Wonka and the golden ticket in his chocolate bar from the children's book, *Charlie and the Chocolate Factory* [89] and the film, *Willy Wonka and the Chocolate Factory* [90]. A female consumer said in reference to the Green & Black's bar:

*. . .the gold packaging with the gold foil is like a Willy Wonka moment. [Chuckling] I don't know. It just feels really special.*

Based upon these focus groups, we propose that for American premium chocolate consumers, in the context of premium or craft chocolate, the gold color on the outside of the package, as the lettering, as the inner packaging foil, or as a theme of the entire package, may indicate high quality and a high price.

To the authors' knowledge, there is no published academic literature wherein the color gold in packaging is linked to the perception of expensive or high-quality products by consumers. In general, there are very few published color-related consumer studies, which is likely because color is highly specific to the type of product being sold, the brand personality desired, and the culture in which the product is being sold [83, 91–94]. Research has shown that package color has a high impact on consumer perception of product quality, brand personality, familiarity, and purchase intent, because color associations are triggered through referential meaning, which happens consciously and subconsciously [92, 95–97]. One way that referential meaning functions is that a strong brand is associated with a distinctive color, which becomes inextricably linked to the product class and is eventually expected to appear in other brands within that product class [98–100]. In our focus groups, the color gold was strongly associated with the fictional character Willy Wonka and the golden ticket he placed in his chocolate bars, as well as high quality and high price associations. Therefore, we propose that Willy Wonka may have been the first to communicate the association of gold and quality chocolate to the generations of children and adults who read the book or watched the movie [89, 90].

**Thick foil indicates high quality.**   In addition to color, consumers found the thickness of the foil to be of tactile and practical importance. A female consumer compared the thin foil of the Lindt bar to the thicker foil of the Green & Black's and Dandelion:

*This one also doesn't rip as easy. Those are, I mean you could just breathe on them and they'll just come apart in your hands. You're like, "Well now I hate to—I have to eat the whole thing." This one is the nice folding ability, really into the foil.*

Consumers fixated on the thickness of the foil as an indicator of quality and lengthy discussions arose in the women's focus groups regarding foil functionality. Thick foil allows consumers to purchase a tablet chocolate bar, eat a small portion, then wrap it up to eat more later without worry that the foil will tear and expose their purses or bags to chocolate pieces.

Previous packaging studies have focused mainly on outer packaging, but Krishna et al. [83] introduced a new taxonomy to packaging, in which they describe an outer, intermediate, and inner packaging. They explain that all three parts of the packaging are important to create a streamlined communication of the brand identity of the product. In our study, the outer chocolate package was either plastic film (Hershey), cardboard (Lindt 70%), or paper (Green & Black's, Endangered Species, Dandelion); the intermediate package was either not present (Hershey) or a silver (Lindt 70%) or gold (Green & Black's, Endangered Species, Dandelion) foil; and the inner packaging was the mold shape used for the chocolate. Our results demonstrate that for premium chocolate consumers, intermediate and outer packaging are of great importance. Interestingly, participants did not discuss characteristics of the chocolate mold itself.

## Credence attribute: Trust

Consumers want to trust the chocolate products they purchase, however, there are a variety of proxies that communicate trust. Some consumers found sustainability labels to be important guides for trustworthiness, while others found sustainability labels to be a reason to distrust a product. Most consumers trusted a chocolate bar that communicated a semblance of "meaning" through a story, promotion of a cause, or a person's name. A male consumer explained his complex rationale for distrusting American government certifications and instead trusting brands:

> *I just still try to make good decisions as far as my health goes, but I'm more concerned with my kid. . .I grew up in Camp Lejeune, North Carolina. . . It's a Marine Corps base. I lost both my parents because the water there was about 20,000 times the legal limit for perc. In my lifetime, the government hasn't really looked out for me that well, so I feel more like it's my responsibility to figure out what I'm putting in my body and the decisions I'm making. I'm not saying regulations are bad; I'm just saying that we do have an awful lot of them, and a lot of times, they're not really getting enforced, and we don't find out until later, or we spend a lot of our tax money or things like the EPA, and they dump all the poison in the Animas River last year. Doesn't always work according to plan. . . . . .That's why. . .I don't know if that Hershey's bar is organic or non-organic, but I know they've been sellin' them for a dang long time. I feel pretty safe buying anything with the Hershey name on it, not that somethin' can't go wrong, but it's like Heinz, or something like that.*

**Sustainability labels as a proxy for trust.** Premium chocolate consumers varied dramatically in their knowledge and importance of sustainability labels, which was directly linked to whether that label served as a proxy for trust. Some consumers knew the exact definition of USDA Organic and fair trade certifications, while others had never heard of the terms before. Still others were confused by the meanings of organic and fair trade and thought they were the same thing.

Some premium chocolate consumers clearly understood Fair Trade and Organic certification and adhered to purchasing only items with these certifications. A male consumer said:

> *Right, organic. It's important to me to have organic produce and organic foods, because I know it hasn't been sprayed with pesticides and chemicals. Fair trade tells me that the*

*purchase of the cocoa beans, they paid a fair price for them, that they haven't taken advantage of the farmer. Those labels are on just about anything I buy.*

A female consumer explained that buying certified chocolate was her compromise for not being able to purchase locally-grown chocolate:

*I think it's appealing, because we don't grow cocoa beans here. You can't, like coffee also. I'm a big coffee drinker. You want to know that, because I can't get it locally, because I want to use it and may as well support something that is at least trying to be environmentally conscious. If they have to put all the little stickers on it, then so be it. I think it's a good thing.*

These consumers were rigid in their belief in sustainability certifications and told us convincingly that they regularly purchased products with sustainability certifications. The NCA survey found that 81% of consumers highly influenced by certifications are willing to pay more for those certifications [9]. Interestingly, the FCIA focus groups found that the importance of sustainability certifications to chocolate enthusiasts varied by US region. Seattle consumers stated that certifications were very important and they would pay more for the them, while only half of participants in San Francisco were motivated to purchase certified chocolates [6].

For the two women participants with adopted children from developing countries as well as other focus group members, fair trade was far more important than organic certification because to them it directly impacted people in developing countries. One of the women said in response to whether she would buy chocolate with sustainability certifications:

*. . .I wouldn't buy just for the organic part of it at all. The fair trade I would. I also would be— I have two children adopted from Guatemala so I know all about the fair trade. Yeah. That would attract me more than the organic. No. I agree [organic is] just a marketing thing. . .*

The NCA survey of American consumers found that fair trade certification is more important to millennials, yet engaging in fair labor practices without mention of the certification is more important to the baby boomer generation [9]. Fair trade certification was found to be more important than an organic label to Flemish and French chocolate consumers [17, 23]. Another Belgian experiment found that consumers preferred the fair trade label over the fair trade and organic label combination, indicating a strong preference for fair trade over organic [101]. In a choice experiment, fair trade chocolate was preferred over non-fair trade chocolate by Belgian chocolate lovers, all things considered equal [18].

Overall, European studies have shown that fair trade certification resonates with consumers because consumers feel strongly about issues related to the exploitation of female and child labor, general working conditions, and human rights [20, 101, 102]. In the NCA survey, cacao farming and chocolate production practices were the least important to mainstream chocolate consumers and the most important to fine chocolate consumers [9]. In our study, some American premium chocolate consumers were concerned about purchasing fair trade products so that producers received a larger portion of the sales.

**Sustainability label confusion.** In addition to consumers with high knowledge and importance of sustainability labels, there were also consumers who were profoundly confused by the label meaning. A female consumer said while eating the Green & Black's chocolate:

*Maybe this is a silly question. What does organic mean here? Is it an ingredient or what? I mean, it's a silly question. I mean, they put it specifically, like after the brand, that it's organic. What does that mean?*

Still other consumers thought that fair trade and organic were essentially the same thing. A female consumer said:

> . . .I know this isn't true, but when I see the word organic, I almost feel like it's synonymous with fair trade.

In the United States, certifications that can be used on chocolate packaging include Fair Trade, USDA Organic, Rainforest Alliance, and Non-GMO Project Verified [103, 104]. In our focus groups, consumers paid the most attention to the USDA Organic and Fair Trade certification logos. The USDA Organic seal is verified by a third party and may only be used if the product meets three main criteria: produced without excluded methods; produced using allowed substances; and overseen by a USDA National Organic Program-authorized certifying agent [105]. The purpose of fair trade certification is to bring transparency to global commodity chains with the overarching goal of transferring capital to producers in developing countries [106]. Fair trade is also a third party certification because the standards are set and implemented by four different organizations [106, 107].

Our findings with American consumers align with European studies that have tested consumers' ability to identify sustainability certification labels. In a Belgian experiment, 60% of Flemish consumers were able to correctly identify the Fair Trade label, whereas only 6% correctly identified the EU Organic label [17]. In the same study, 16% of Flemish consumers thought that organic chocolate used fair trade cocoa, while 11% of consumers thought that fair trade chocolate caused less pollution, and only 20% of consumers correctly indicated that fair trade chocolate uses sustainably produced cocoa [17]. In a survey with Italian consumers, 44% ticked the box, "I find it difficult to interpret the information on the label" when looking at Fair Trade, Rainforest Alliance, and $CO_2$ reduction labels [20]. A study of 126 Brazilian consumers found that 73% wanted their chocolate labeled with certifications and 79% said they would pay more for it, yet few recognized the certification seals: only 56% correctly identified organic, 36.5% for origin, 15.9% for Rainforest Alliance [85]. A survey comparing German and UK sustainable consumers found that nearly all UK consumers are familiar with the fair trade label, compared to 90.3% of German consumers [24]. In the same study, 41% of UK consumers were familiar with the organic label, compared to 97% of German consumers [24]. Sustainability label consumer confusion appears to be an international issue and American craft chocolate makers have reacted to this by rejecting the use of sustainability labels and instead opting to use direct trade systems they organize themselves [5].

**Sustainability labels are not a proxy for trust.** For some consumers, fair trade and organic labels were not important and instead consumers were neutral to them. As one male consumer put it:

> If I like the chocolate, and I can afford the chocolate, I'll buy the chocolate. I think it's great to —this is all political to me, and I'm not naturally a huge political person. I care about the world. I care about humanity, generally, [laughter] but when I go to buy a chocolate bar, I honestly just don't think about it. I don't think about this stuff.

For other consumers, the sustainability labels raised red flags and they were skeptical of greenwashing [108]. These consumers were not sure if they could trust that a company would certify for several labels and execute them all in a meaningful way. A male consumer:

> Yeah, I'm questioning, are they just trying to sell their chocolate, or are they actually supporting all these, and if they're supporting, is it in a good way? I really have no idea, and that

*would take a lot of research to figure that out. I'm just trying to get a chocolate bar is my bottom line, so I question all of that right now, to be honest. I don't know.*

Rousseau's [17] survey and choice experiment found that a majority of Flemish chocolate consumers were indifferent to organic certification and they were willing to pay less for chocolate labeled organic than chocolate with no organic certification. Ultimately, 42.9% of study participants believed organic certification was a marketing tool, while 32.5% thought that fair trade was a marketing tool [17]. Rousseau [17] concluded that consumer willingness-to-pay for these certifications is strongly linked to the product being considered, the country, and region. Additionally, consumers who eat chocolate frequently were found to value fair trade and organic certification more than infrequent chocolate eaters [17].

The FCIA work found that for chocolate enthusiasts and connoisseurs, certified fair trade is more important than direct trade, however, any certification would not impact their willingness to purchase chocolate [6]. One of the six segments found in the American Midwestern millennial candy bar study was the anti-organic group, which composed 11% of the sample [27]. They had a strong preference for high fat chocolate, no preference for "clean" labels, and a distinctive dislike and distrust for organic and non-GMO products [27]. Our finding of sustainability certification neutrality in chocolate products aligns with Belgian and American research, which may be because consumers feel most strongly about sustainability certifications on fresh produce rather than chocolate [109].

**Price and flavor are more important than sustainability certifications.**  Several consumers expressed that they prioritize price and flavor ahead of sustainability certifications in chocolate purchase scenarios. A female consumer explained:

*Price is also another thing. . .especially on a college budget, organic a lot of times is way more expensive. You say, "Whoa. What the heck? It tastes kinda the same to me." I don't think there's a difference and it's really much more expensive. All right I'm just gonna go with the other one. If I really can't taste the difference. All right we're just gonna go with the other one.*

One male consumer stated when asked about purchasing chocolate with sustainability certifications:

*Sometimes, but when I do, it tends to be more a decision more about—that I've compared it to some other foods and found the quality to be better. I buy yogurt pretty often, Greek yogurt, and that there's a—I think Stonyfield makes some organic Greek yogurt that it just tastes really good compared to the Giant store brand Greek yogurt, so I buy it that way, and I guess some—it's nice that it's organic, but the primary motivation is that it's simply just really good.*

Our findings are consistent with several choice experiments from which "willingness to pay" and "willingness to buy" values are calculated, which have found price to be a significant barrier to purchasing products with sustainability certifications [110–115]. French chocolate consumers participated in a Becker, DeGroot and Marschak auction that later segmented them into three groups: 42% of the sample was very sensitive to price and the least sensitive to organic and fair trade labels; 41% of the sample unconditionally adhered to organic and fair trade labels and their willingness to pay was very high; 17% of the sample had the highest willingness to pay and only purchased organic and fair trade subject to the taste of the product [23]. Similar results were found in a choice experiment with UK and German chocolate consumers, where in comparing various levels of price, cocoa origin country, country of chocolate manufacture and sustainability labels, nearly 50% of both groups of consumers selected price

as the most important attribute in their purchase decisions [24]. Surprisingly, even when English consumers were informed of the injustices in the cocoa supply chain, they admitted that they were unwilling to pay a price premium of 10–15% for chocolate if it was produced in a more socially responsible way [116]. In a choice experiment, Belgian chocolate consumers positively valued both taste and ethical considerations, but taste proxies such as product type (boxed chocolate or chocolate bar), chocolate type (white, milk or dark), or praline filling (alcohol) ultimately dominated their decision making [18]. In the FCIA research, chocolate enthusiasts said that taste was the most significant reason to purchase craft chocolate [6].

Ultimately, the most important indicator of whether a premium chocolate consumer found chocolate products with sustainability certifications desirable was what the sustainability label meant to the consumer. In our study, sustainability label meaning and importance appear to be in a 2X2 matrix, with importance on the x-axis and meaning on the y-axis. If the label means safe or fair, and the consumer values safe or fair, then it is important. If it does not mean safe or fair, and the consumer values safe or fair, then the label is unimportant, and the consumer is unlikely to purchase it. This meaning and importance matrix is consistent with Carrigan and Attila's "Consumer attitudes to ethical purchasing" 2X2 matrix developed from their focus group work in the UK, in which they identified four types of consumers: Caring and Ethical, Confused and Uncertain, Cynical, and Disinterested and Oblivious [116].

Other researchers have proposed consumer decision making models of sustainably-certi-fied products, however, none of these models incorporate trust [102, 115, 117–119]. Our find-ings made clear that trust or mistrust of sustainability certifications are important credence attributes for premium chocolate consumers, which was also reported for Italian chocolate consumers [20].

**Finding "meaning" in a product.** Whether sustainability labels were important to them or not, consumers searched for a semblance of purpose in their chocolate bars. The "meaning" focus group participants described is what Pink refers to as the "integration of meaning" [120, 121]. Modern consumers look beyond traditional marketing techniques and search for mean-ing through a story, shared values, and committed personal interaction [121].

Focus group participants were excited by packages that told stories outright, such as the Endangered Species chocolate bar, which has an image of an endangered animal on the front and a story about the animal on the inner label of the bar (see Image 2 https://doi.org/10.26207/a863-pp02). In addition to being told a story, consumers were also excited to imagine stories about the people listed on the Dandelion bar or create ideas about the chef pictured on the back of the Lindt chocolate bar. A male consumer said:

*These ones that we're looking at, I find it interesting because it's almost like a story to be told, some type of description on 'em, versus you get a Snicker's or a Reese's, I mean it's just a Snick-er's or Reese's.*

Storytelling marketing or "content marketing" has existed as a concept for more than 30 years [122]. This type of marketing applies well to craft chocolate wherein a story can be about how cocoa beans were sourced or how the chocolate itself is made. In her work, Leissle found a story to be part of the artisan chocolate construct because artisan chocolate companies typi-cally have a person who can share the story with the consumer, which ultimately forms a con-nection [4].

Focus group participants expressed their strong desire to purchase chocolates that benefit a cause because it allowed them to share their values with the company and reduce guilty feel-ings that they had for indulging in chocolate. A female consumer explained:

*I think it would be the donation portion or the donation part of it that would drive me to that one rather than—I don't necessarily care about organic. . .I don't care about gluten-free. GMO, Eh, whatever. [Chuckles] I do like the aspect of, Well, okay, and it sounds silly, but I'm buying this chocolate and I'm doing something. It makes you feel good. That's probably why I would pick that over—*

"Cause-based marketing" was conceptualized over 30 years ago to describe the linkage of fund raising for the benefit of a cause to the purchase of a firm's products [123]. This is likely explained by the fact that ethical consumption, the act of using ones political, spiritual, environmental, social or other motives for selecting one product over another, has been on the rise for the past 50 years [124]. Langen [119] used a choice experiment to investigate how consumers perceived giving donations as part of a coffee purchase. In the study, 70% of respondents had a strong preference for fair trade and organic production, but disliked cause-related marketing, while 27% of participants preferred cause-related coffee to sustainability certifications [119]. This 27% of participants had a significant and positive willingness-to-pay of 0.55 euro to support a cause [119]. In contrast, nearly all of the premium chocolate focus group consumers demonstrated high interest in purchasing chocolate for a cause and were particularly excited to support endangered species.

Focus group consumers expressed guilty feelings about purchasing an indulgent chocolate bar for themselves. We propose that purchasing chocolate that benefits a cause reduced this guilt in a process known as guilt appeal [125]. Previous work has suggested that cause-related marketing works best for products that are perceived as frivolous, such as chocolate, rather than products that are a need, like detergent [126].

Focus group participants found meaning in the names written on the chocolate bars as either the brand (e.g. Hershey's), the cocoa sourcer (e.g. Greg), or cocoa roaster (e.g. Chiann) (see Image 2 https://doi.org/10.26207/a863-pp02). The name "Greg" is listed on the back package of the Dandelion chocolate bar and focus group participants crafted positive stories about Greg. In lieu of an actual personal human interaction, this mark of a name represented a personal connection and effectively communicates everything that Dandelion wants to convey about their size and handcraftedness [4]. A female consumer explained:

*I was assuming that Greg is the grower, picker, supplier of these cocoa beans that make that chocolate bar and possibly his sugar plantation. I'm not sure. . .but I was gonna say it's kind of bethel. . .it makes it sound. . .like it's a craft, like more of a small operation, I picture Greg with his stuff, his plants, and making chocolate. . .I would think of it more like a labor of love, making these delicious chocolates.*

The Hershey brand name communicates Hershey company values [127]. A male consumer stated:

*I don't know Milton Hershey personally, but I know he built that school with all that money. I've got close personal friends that went to that school, and they get their education paid for life, so just how horrible is Milton Hershey? Is he in the Dominican Republic abusing people making them? I don't know. I tend to think not. . .*

The association of a person's name with a chocolate bar or using a person's name as a brand, helps to form a brand personality [91]. We postulate that this set of human characteristics associated with a brand is part of the personal affect that creates a relationship between the consumer and the chocolate bar [128]. As the relationship continues, brand loyalty is achieved

and ultimately, brand love [129, 130]. Nearly 40% of Fortune 500 brands use a person or place name to help create a relationship, so this phenomenon is not unique to chocolate [131]. However, our focus group results suggest that in the context of chocolate, a brand name reassures consumers that what they are buying is trustworthy and was made by a person.

Focus group participants created meaning and a personal affect by crafting stories about Greg and his chocolate sourcing. In so doing, they picked up on the three main attributes of the Dandelion brand: intimate, craftsman, and exquisite [2]. In work with Costa Rican cocoa production, supply chain management, and marketing, Haynes et al. [132] found that integrating meaning into cocoa and chocolate products could help chocolate companies and products stand apart. They viewed integration of meaning as an achievable approach for agile small producers and chocolate companies, and a method that effectively communicates what certifications attempt to communicate, without the drawbacks of expense and confusion [132]. The concept of meaning aligns with Leissle's artisan concept because both attribute a person behind to the company [4]. Research has revealed that consumers drink craft beer for the meaning it conveys, which much like craft chocolate, allows a consumer to build a unique identity, in comparison to mainstream industrial beer or chocolate consumption [4,82].

Trust is an important attribute of a chocolate bar for premium chocolate consumers. Trust is mentioned frequently in consumer literature and is defined as "a particular level of the subjective probability with which an agent assesses that another agent or group of agents will perform a particular action" [133]. Gambetta states that when we say we trust a product, "we implicitly mean that the probability that [it] will perform an action that is beneficial or at least not detrimental to us is high enough for us to consider engaging in some form of cooperation with [it]" [133]. Focus group participants wanted to trust the chocolate bar they were consuming and discussed sustainability labels or an absence of sustainability labels as a form of trust. Whether consumers found sustainability labels to be trust proxies or not, most sought to find meaning in their chocolate products, in the form of a story, a cause, or a personal affect.

## Experience attributes: Joy and/or utility

In addition to trust, consumers look for experience attributes, utility and/or joy, in their chocolate. We believe that joy and utility may have a cyclical relationship that depends on the consumer need state and at some point, the lines may become blurred.

**Chocolate is linked to holidays, celebrations, and gifts.** Consumers associate chocolate with specific holidays, such as Halloween, Easter, Christmas, and other celebration and gift giving. As a male consumer explained:

*. . .chocolate and sweets in general are tied to some of my most favorite memories of my life, even growing up, because they were always tied in conjunction with, usually, a holiday, or a ceremony, or a specific event. . .. these amazing memories of growing up, it was connected with it almost exclusively, outside of any other food, outside of maybe turkey for Thanksgiving.*

Chocolate, holidays, celebrations, and gifts go hand-in-hand internationally. Italian chocolate consumers strongly associate chocolate with Easter and Valentine's Day [134]. Focus groups with French and American (New York and Pennsylvania) chocolate consumers showed that French consumers use chocolate gifts to affirm social ties during celebration, gratitude, or loss, while Americans also give chocolate gifts, but less readily, prioritizing cost, presentation, and brands [88]. Additionally, a majority of the American focus group participants linked chocolate to childhood and shared their memories of eating chocolate in birthday cakes, cookies, candy bars or ice cream at holidays [88].

**Nostalgia through packaging, flavor, taste, and mouthfeel.** Our work suggests that because children consume chocolate at a young age and it is associated with mostly positive events, chocolate may become associated with positive feelings. Throughout a lifetime, as children grow into adults with purchasing power, they may seek out chocolate attributes that remind them of their youth or nostalgia. Nostalgia is defined as "a preference toward objects that were more common when one was younger" [135]. These nostalgic attributes come from packaging, flavor, taste, and mouthfeel. Several participants associated chocolates with Grandma's house in the Special Occasion section of their product maps (Fig 2). A female consumer explained:

*Just to go back really quickly to the idea of the box of chocolates. . .I get a little nostalgic at least with the Whitman samplers. Because that was what my grandmother would get for everything. We'd go to grandma's and she'd have the box there. . .There's sometimes when I'm feeling like particularly nostalgic or I want a little—I'll buy one of those, the small boxes, the ones—because—but you know what you're gonna get in that box.*

Focus group participants discussed at length how the Hershey bar of their youth had an inner-foil package that was wrapped with a brown paper label. When sampling the Hershey bar, a female consumer said:

*You know it's a Hershey bar. You asked a little bit about the packaging. Well, I remember when the packaging, it used to come in—it was paper and then there was foil and you had to unwrap it. I mean, I know this is the way things are going and obviously it's cheaper and everything, but I wish—the packaging just seems—it's not as big a deal as opening a Hershey bar, as it used to be. . .Yeah. It's like Charlie and the Chocolate Factory, 'cause you had to open the paper and you peeled back the foil, and, oh, there's the golden ticket*

When consumers thought of the special two-part process of opening the Hershey bar of their childhood, once again, they thought of Willy Wonka [89, 90]. Our results may suggest that because Willy Wonka is introduced when children are at a young age, through both a book and film, it may be embedded with positive, nostalgic emotions that carry on through adulthood.

Nostalgia was not only important to packaging, but also flavor. A female consumer explained while eating the Hershey bar:

*I just feel like it's such a basic bar, but satisfies so many of my needs. The cocoa just lingers in my mouth. It's the perfect amount of sweetness. Not too sweet especially with the morning or in the afternoon. Just the taste associates with childhood memories and s'mores or chocolate covered pretzels, or that whole thing I do when I was a kid. . .*

Research with Italian consumers revealed that some of the main chocolate thoughts in diaries of "chocolate lovers" were nostalgic. Participants waxed poetic about eating Nutella and associated it with their youth [134]. Australian chocolate consumers revealed that Cadbury made them feel nostalgic [80]. Based upon our findings, Nutella is to Italian chocolate lovers as Cadbury is to Australian chocolate consumers and Hershey bars are to American premium chocolate consumers.

Chocolate is a unique specialty food because unlike beer or coffee, it is consumed from a young age onwards. Our focus group discussions revealed that when that child becomes an adult, their tastes may mature from milk to dark chocolate, or even from Hershey's to Lindt to

Dandelion chocolate. However, as Mintel consumer research confirmed, this same consumer would likely still eat milk chocolate to remember their youth or grandma's house [16]. We propose that there may be a taste progression for some consumers and others may never enjoy eating dark chocolate. A male consumer explained:

> You know what I notice is you know how kids—grow up drinkin' grape drink instead of grape juice, or American processed cheese instead of cheese? Then, they think that's what cheese tastes like, or that's what grape tastes like, whatever. . .I think that beer, and bread, and chocolate are really very similar.

**Consumers differentiate between chocolates they buy as gifts and chocolates they buy for themselves.**   Consumers strongly differentiated between the types of chocolates they buy as gifts and chocolates they buy for themselves. In general, they gifted higher quality chocolates than they would purchase for themselves. A male consumer said:

> When I think quality, that would be something I would buy for somebody, maybe, because it was. . .something different, something interesting. . .

This idea was reiterated by Finnish chocolate consumers, even heavy store-brand purchasers, who stated they would never purchase a store-brand chocolate as a gift [79]. Giftable chocolates, which may align with the NCA's premium chocolate or participants' special occasion segment, are considered a unique segment for premium chocolate consumers.

Focus group participants had different classes of chocolates for different purchase occasions and some admitted to "treating themselves" by purchasing special chocolates. A male consumer explained his purchase:

> There's a chocolatier in Philadelphia, John and Kira's. Lovely chocolates, but they're really, really expensive. They end up being $5.00 and $6.00 per piece, so they're very pricey, handmade chocolates. Those are the ones I tend to buy, because I don't eat a whole lot. A 50-pound box of chocolates from John and Kira's may last me six months, because I just won't sit and eat. I just, "I'm gonna treat myself tonight, and I'm gonna have a piece of that chocolate," but I don't tend to buy the bars in the stores because I would consume all of it. It's true.

Our findings align with the 2017 Mintel Snacking in Foodservice Report, in which the number one reason that 54% of consumers gave for eating snacks was "to treat myself" [136]. This is similar to much of the coffee consumer literature, which has found that coffee is consumed as a personal moment of pleasure [137–139].

**Chocolate is consumed for desirable post-ingestive effects.**   Focus group participants oscillated between purchasing chocolate for joy and using chocolate as a utility for its desirable post-ingestive effects. Participants described using chocolate to energize, particularly when driving. A male consumer stated:

> . . .when I'm traveling, I usually need to have some type of chocolate and some water or something like that, just to kind of power up.

Consumers commonly associate products with energized feelings. Coffee consumers have reported purchasing and consuming coffee because of its mentally and physically stimulating properties [140]. Italian consumers stated that chocolate gave them energy when they were tired, after exercise, or strenuous mental activities [134]. Focus group participants use of

chocolate for energy lines up with scientific evidence, which shows that chocolate contains psycho-pharmacologically active compounds [141].

In the women's focus groups, much discussion revolved around craving chocolate during perimenstruation. Women described eating higher amounts of chocolate or different types of chocolate to ease their discomfort. A female consumer said:

*I think maybe there are certain times of the month that I would prefer a Hershey bar than I would prefer the dark chocolate.*

Chocolate is the most commonly craved food in North America [142]. However, its use as an aid during perimenstruation is unique to North American women as a study comparing Spanish and American women found [143]. Scientific studies have shown that while there is no physiological rationale for consuming chocolate to quell perimenstrual distress, many American women use perimenstrual syndrome (PMS) symptoms, as a culturally acceptable reason to indulge in chocolate [39–41].

Consumers also linked chocolate to relaxation or a quiet moment. As a female consumer noted:

*There are just some mornings where it's a little more stressful than other mornings and it's like that chocolate just gives you that, "Ah. Okay I'm ready to go now." . . .Yeah there are times where yes mom needs chocolate in the morning. [Laughter] Just depends.*

Similar to our findings, Italian consumers described keeping chocolate for themselves to enjoy alone for a moment of escape [134].

Focus group participants described several uses for chocolate (energy, PMS discomfort, relaxation) with language that made chocolate sound medicinal. At some points of the conversation, participants described a "desperate need" for chocolate as though there was a deficit that could only be satisfied by chocolate. One female consumer said:

*. . .I have been known, in desperation, to drink Hershey's syrup out of the container. [Chuckles] This is the only group I've ever said that to. . . [It's an] emergency bottle that you can— [chuckles]. You don't have to bother chewing.*

Another female consumer explained, referring to her product map:

*It is interesting, because I think that, depending on the mood that you're in or your need for chocolate, can affect how you group. I mean, if you're in desperate need, then it's all in one big pile.*

Focus group participants frequently explained their chocolate need state as a "desperate need". This is consistent with other Pennsylvania chocolate consumer focus groups findings, in which one participant depicted herself as a crazy woman who needed a chocolate "fix" whenever she was stressed [88]. A "desperate need" was also described by Italian consumers who wrote diary entries about chocoholism, a physiological and psychological dependence on chocolate [134]. Rogers and Smit [144] determined that addiction to chocolate is more accurately defined as a craving and propose that people are ambivalent about consuming chocolate because there is a cultural norm that chocolate is highly palatable but should be eaten with restraint. When the concept of restraint is pitted against a human desire to consume, craving results [144].

Zarantello & Luomala [134] identified four rationales for Italian chocolate consumption: medicine (physiological or sensorial need), mind maneuvering (escapism, nostalgia), regression (materialism), or ritual enhancement (interpersonal gifts) [134]. This finding aligns with our finding that premium chocolate consumers oscillate between purchasing chocolate for utility and/or joy purposes.

Our findings are consistent with the 33% of consumers in the NCA survey who ranked mood as the number one driver influencing their chocolate purchase and FCIA survey participants for whom pleasure was the biggest motivator to purchase fine chocolate [6, 9]. In focus groups with American consumers, Terrio [88] also found that Americans derived pleasure from eating chocolate for more energy, less stress, happier moods, and to relax. The relationship between mood and chocolate is strong and such that one could be in a good mood and buy chocolate as a treat or be in a bad mood and attempt to reinstate a good mood through chocolate.

## Study implications

**Methodological implications.**   Our work demonstrated the efficacy of using projective mapping and focus groups in tandem to produce rich findings on the exploratory topic of premium chocolate consumer perception of craft chocolate and chocolate quality attributes. We used individual product maps as a "show-and-tell" object at the beginning of the focus group and it provided a common ground for all participants to share how they define and separate chocolate products. This activity was an excellent way to spark discussion about chocolate, provided an ice breaker to decrease the inherent awkwardness that comes from talking with strangers, and also enabled us to get straight to the discussion. Risvik [29] was correct to suggest that a projective mapping activity could be done prior to, and then used in a focus group to enhance the conversation. The follow-up analysis of the individual product maps using Multiple Factor Analysis created one product map and visually displayed how premium chocolate consumers separated their products. This proved to be an effective non-verbal technique to understand how consumers perceive and segment chocolate products based upon quality in an unbiased setting.

On its own, projective mapping does not provide nuanced information about how and why consumers segmented products. Focus group data, when analyzed using grounded theory, provided deeper information about premium chocolate consumer perception of craft chocolate and desirable chocolate product attributes. However, robust focus group data analysis takes a tremendous amount of time and thus there is a trade-off in the richness of data collected and the time it takes to collect and analyze it.

Using both methods in tandem allowed us to develop a rich picture of how premium chocolate consumers judge quality and perceive craft chocolate. Esmerino et al. [145] also conducted focus groups and did a projective mapping activity with fermented milks. They concluded that projective mapping data was high-quality and quick to collect and analyze, which was preferable to the focus groups, which took a long time to conduct and analyze [145]. In contrast, we found that both methods performed in tandem provided the richest data possible, particularly for a novel and understudied food product such as craft chocolate. This method is recommended for future exploratory research.

**Commercial implications.**   This paper presents the novel finding that consumers segment products differently than the NCA when given only packaging information (without pricing). The NCA may want to re-evaluate their segmentation method and consider the utility of chocolate products, in addition to price, in their product segmentation. Additionally, premium chocolate consumers clustered premium chocolate with craft chocolate, showing that in the

absence of price information, they do not differentiate premium chocolate bars and craft chocolate bars. We recommend that craft chocolate makers investigate methods to distinguish themselves from premium chocolate to justify their price and encourage premium chocolate consumers to trade up. One method could be to use premium chocolate consumer descriptors of craft chocolate, such as those used during the craft chocolate sampling, to emphasize flavor and taste attributes when communicating their product. Additionally, we recommend that craft chocolate companies acknowledge the extrinsic packaging details that premium chocolate consumers paid most attention to: chocolate origin, a handcrafted aesthetic, cocoa percentage, the color gold, and thick foil.

Ultimately, consumers desire trust as a credence attribute and joy and/or utility as experience attributes. There is no clear guidance for the use of sustainability certifications because consumer knowledge, perception, and consequently, trust varied greatly. However, all consumers in our focus groups were interested in meaning, which craft chocolate makers can communicate through the use of a story, a cause or a personal affect. Once craft chocolate companies succeed in communicating trust to the consumer, they may directly target other elements of joy, such as nostalgia, and/or utility, such as gifts.

**Study limitations.**   While effective, this study had a few limitations. One limitation was that we were limited to a sample population from a university town in Central Pennsylvania. A more robust sample of different geographical locations in the United States may have yielded different results. However, we are satisfied with the convergent validity of our findings with the results of the FCIA and NCA research which had a broader, national sample population [6, 9].

The projective mapping activity used 47 chocolate product stickers and pieces of paper. Due to the size of the stickers, information about price, ingredients, and weight could not be included and some of the stickers were lost by participants and had to be replaced. Additionally, some of the 47 stickers overlapped in the same categories, indicating that the same findings may be possible with fewer stickers. A maximum of five stickers per chocolate segment, for a total of 15 stickers is recommended for future studies. Another limitation was the use of paper and stickers in place of computer software. While paper and stickers allowed for participants to touch each sticker, put the sticker in the correct location on the map, and use their maps during the focus groups, the measurement of X, Y-coordinates and word descriptors was time consuming. Future studies could use software on tablets to perform the projective mapping activity so that participants could show the tablet during the focus group and then the software could tabulate product distances and group word descriptors instantaneously.

Focus group participants sampled five chocolate bars selected to represent mainstream, premium, and craft chocolate, and each group ate the five chocolate bars in the same order. The results of this study may be specific to these chocolate bars and consumption order, which may have been different had we used different chocolate bars and randomized the order. Future studies could use different chocolate bars and present them in a randomized order, although this would require several more focus groups to insure reproducibility. A final limitation was the slow rate of focus group data analysis using the "scissor and sort" technique. Future studies could use computer software for coding instead.

The main limitations of focus groups are social desirability bias and also the attitude-behavior gap. In our study, we believe we minimized social desirability bias by having consumers create their projective map beforehand. However, it is difficult to overcome the attitude-behavior gap in consumer research in general.

**Suggestions for future research.**   This study is one of the only consumer perception studies with American chocolate consumers. More studies could be conducted with a broader geographical variety or type (mainstream, premium, craft) of American chocolate consumers.

This study suggests that there may be gender differences in consumer use and perception of chocolate products. Future work could delve deeper into isolating these differences and understanding how to market chocolate to each gender more effectively.

To target specific search, credence, or experience attributes of craft chocolate, future studies could be choice experiments or experimental auctions where consumers must make a choice as to what product they prefer in a realistic environment. This would allow for search, credence and experience attributes highlighted in this study, such as cocoa content, price, brand names, and sustainability certifications to be pitted against one another and determine what a consumer would ultimately select in the marketplace and their willingness to pay for it.

## Conclusion

This study demonstrated the utility in using both qualitative (focus group) and quantitative (projective mapping) techniques to uncover consumer perception of a novel product such as craft chocolate. Through this work we revealed that premium chocolate consumers segment chocolate products differently than the NCA and form a unique segment for special occasion chocolates. When premium chocolate consumers tasted craft chocolate, they expressed excitement, were intrigued by the flavor, and had difficulty contextualizing it compared to the premium and mainstream chocolates. Premium chocolate consumers paid close attention to search attributes such as a handcrafted aesthetic, the color gold, cocoa percentage, and thick foil in all of the products they mapped and sampled. Grounded theory analysis of focus groups revealed that in addition premium chocolate consumers find trust to be the primary desirable credence attribute, while utility and/or joy are primary desirable experience attributes.

## Supporting information

**S1 Appendix. Focus group moderators guide.**
(PDF)

## Acknowledgments

The authors would like to thank our participants for their time and effort. We also thank Andrew R. Cotter for plotting coordinates, Patrick J. Dolan and Jacob S. Ginn for their help in projective mapping coding, Marielle J. Todd and Dr. Weslie Khoo for feedback on the conceptual map and image editing, John Russell, Dr. Heather Froehlich, and Dr. Stéfan Sinclair for their advice with word analysis, and Dr. Thierry Worch for consultation with projective mapping data analysis.

## Author Contributions

**Conceptualization:** Allison L. Brown, Alyssa J. Bakke, Helene Hopfer.

**Formal analysis:** Allison L. Brown.

**Funding acquisition:** Helene Hopfer.

**Investigation:** Allison L. Brown.

**Methodology:** Allison L. Brown, Alyssa J. Bakke, Helene Hopfer.

**Resources:** Helene Hopfer.

**Supervision:** Helene Hopfer.

**Writing – original draft:** Allison L. Brown.

**Writing – review & editing:** Allison L. Brown, Alyssa J. Bakke, Helene Hopfer.

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
