## [Decision Letter · Decision Letter 0]

10 Jun 2020

PONE-D-20-12240

“I’ve got the golden ticket!” Understanding American premium chocolate consumer perception of craft chocolate and chocolate quality using focus groups and projective mapping

PLOS ONE

Dear Dr. Hopfer

Thank you for submitting your manuscript to PLOS ONE. After careful consideration, we feel that it has merit but does not fully meet PLOS ONE’s publication criteria as it currently stands. Therefore, we invite you to submit a revised version of the manuscript that addresses the points raised during the review process.

Please rewrite the paper taking into consideration the **numerous limits** indicated by reviewers.

We look forward to receiving your revised manuscript.

Kind regards,

Patrizia Restani, Ph.D.

Academic Editor

PLOS ONE

Journal Requirements:

'A.L.B. declares no competing interest. We have read the journal's policy and the

authors of this manuscript have the following competing interests. A.J.B. has been

employed in the food industry. A.J.B. and H.H. have received consulting fees from

corporate clients in the food industry. Additionally, the Sensory Evaluation Center

(SEC) at Penn State conducts routine consumer tests for the food, packaging and

packaged goods industries to facilitate experiential learning for undergraduate and

graduate students. All authors have professional relationships with members of the

chocolate industry. None of these organizations nor the funders have had any role in

the design of the study; in the collection, analyses, or interpretation of data; in the

writing of the manuscript, or in the decision to publish the results.'

Additional Editor Comments (if provided):

The paper is interesting and certainly appealing, but the concerns raised by the two reviewers need a thorough review. In particular, it is necessary to eliminate the "promotional" aspect towards a specific product, as well as it is essential to give the right weight to the very limited group of enrolled consumers. The paper requires an important improvement in drafting but also in defining the limits of the study correctly.

Reviewers' comments:

Reviewer's Responses to Questions

**Comments to the Author**

1. Is the manuscript technically sound, and do the data support the conclusions?

Reviewer #1: No

Reviewer #2: Yes

2. Has the statistical analysis been performed appropriately and rigorously? 

Reviewer #1: N/A

Reviewer #2: Yes

3. Have the authors made all data underlying the findings in their manuscript fully available?

Reviewer #1: Yes

Reviewer #2: Yes

4. Is the manuscript presented in an intelligible fashion and written in standard English?

Reviewer #1: Yes

Reviewer #2: Yes

5. Review Comments to the Author

Reviewer #1: This is a very interesting paper that contributes to understand, using an innovative method of analysis, which are the qualitative and quantitative parameters guiding the choices of a part of American craft chocolate consumers. However, it should be noticed that the number of consumers selected is very small (only 27 subjects) and that the group was chosen in a restricted geographical area, that cannot represent the “American consumers” in general. This aspect has been reported in the “limits of the study” section, but, for this reason, the title should be modified because it suggests a larger sample including different geographic areas. Furthermore, even though very appealing, the title reports the “golden chocolate coating sheet” which is referred to a specific brand in the manuscript.

Some other aspects need modifications and/or more explanations:

Line 55: The authors reported only two trade names as an example of "gourmet" chocolate. This aspect seems to be unfair from a commercial and marketing point of view. It would be better to report a table or a link to a website (if available) where the different categories and trade names are listed. The same observation can be done for “Dandelion Chocolate,” Line 57.

Line 61-65: On the basis of which criteria these commercial examples have been chosen?

Line 201: The consumption of chocolate "two to three times per month” doesn’t be considered a frequent consumption. Please explain.

Line 202-203: See comments reported above for the statements at Line 55.

Line 204: How many subjects have been initially screened? Please add this information in the text.

Line 248: Please explain with more details why these brands were chosen in the study.

Line 308: Please briefly explain the "scissor-and-sort" approach.

Line 321: The description of participants’ reaction is reported only for Dandelion chocolate; it should be described for the other categories as well.

Line 454-455: The specific marketplace names cited by a participant should not be reported as have been correlated with chocolate quality.

Fig.3 should be integrated with the attributes used by consumers to define the quality "Research question 2".

Minor typing errors: please correct "two listserves" at line 195.

Reviewer #2: General comments

The structure of the paper should be improved. The text is quite long and readability would improve from writing more concisely. I would also strongly recommend to include more subtitles and to focus on highlighting clear take-away messages for the reader. For example, the section on ‘packaging’ is six pages of continuous text (p26-32), while the section on ‘trust’ is even double that (p32-p43). Subtitles are needed.

It would also interesting to start the result section with an overview of the main elements addressed by the participants and I think these main elements can be categorized in two (or more) categories. From my reading, it may be interesting to distinguish objective (cacao percentage, price, packaging…) vs subjective (trust, joy, perceived sustainability…) product characteristics. Another possible distinction that may help to structure the participants’ arguments is between search, experience and credence characteristics. See, for an example including an experiment with a chocolate bar: Wright, A.A., & Lynch Jr, J.G. (1995). Communication effects of advertising versus direct experience when both search and experience attributes are present. Journal of Consumer Research, 21(4), 708-718.

I am especially concerned about the impact of order effects on the results. Each group was confronted with five types of chocolate in a fixed order. However, this order has (very likely) an impact on the product attributes that respondents discuss. Suppose the first type of chocolate inspires a discussion on, for example, taste and nostalgia. When the second type is presented, participants may feel that taste has already been discussed in details and thus that they should focus on the next topic e.g. the packaging. This would lead to a discussion of more obvious characteristics for the first types of chocolate and of secondary characteristics for the types presented later. It would thus be advisable to randomly vary the order of the chocolate over the focus groups.

I am also concerned about the impact of having one of the five chocolate bars that is ‘new’ to the participants (the Dandelion bar), while the others were already presented in the projective mapping exercise. Now an additional factor, familiar vs new, is introduced into the study which makes the interpretation of the results more challenging.

The impact of gender on the discussion should also be presented differently. The authors assume that there will be a difference between men and women when it comes to the appreciation of premium chocolate. However, the contribution of the study would be improved if one or two mixed (half male, half female) groups would have been included. Also, it would be interesting to comment on the presence or absence of gender effects in the projective mapping activity.

Past studies have shown that occasional and frequent users of certain products behave differently and focus on different characteristics. Why does the current study only focus on frequent users? The market share of occasional chocolate consumers may be equally larger or even larger. The focus on frequent users may also introduce a type of bias to the results. As they are selected because they are ’chocolate experts’, they may feel a drive to show off their expertise and to demonstrate the variety of their knowledge.

Detailed comments

- I would recommend to shorten the title as it is too long, in my opinion. A title such as “Understanding American premium chocolate consumer perception of craft chocolate and chocolate quality” is more than adequate.

- Abstract – Line 489: What is a ‘European aesthetic’? How does it differ from an ‘American aesthetic’?

- Mention in the introduction that the study deals with the US market.

- Line 33: 129 craft chocolate makers in the US? In the world?

- Line 45: why is craft chocolate dependent on time? Why only after 1996?

- Line 52: change verb tense in text: the present tense is used while the text discusses old definitions that have been adapted.

- Line 173: please rephrase – focus groups are not ‘the only suitable method’. A series of non-standardize in-depth interviews may also be a valid approach.

- Line 183: mention that the mapping of chocolate relates to quality. Motivate this choice.

- Line 263: which version was used in the study? Why was the label changed? How?

- Line 271-272: the text mention that four focus groups made ‘it possible to reach the point of theoretical saturation’. Did this in fact happen in the current study?

- Line 291: explain first why only 24 maps were used and not 27.

- Line 293: why were three maps ‘unusable’?

- Line 334: the reference to “T.J.” “Maxx” was not clear to me. As a European I am not familiar with American stores. Maybe explain briefly.

- Line 337: the mention of the name “Greg” was very puzzling at this point. Consider reframing.

- Line 429: the observation that higher prices are associated with higher quality levels off for wine. So maybe for chocolate as well.

- Line 566: the link between color cold and expensive, high-quality product is very likely to be context dependent (product, cultural, time…) and may be influenced by the biases in this study. Thus this type of statements should be phrased carefully.

- Line 698: not sure if this study can be labeled as a ‘forced’ DCE since an opt-out was included.

- Line 725: Link to greenwashing may be made here

- Line 750: Why not start a new section here? Because it reads a discussion.

- Line 1149: DCE still suffer from attitude-behavior gap issues - especially price is not considered correctly by respondents, but also product availability and the selection of attributes in the study matter.

6. PLOS authors have the option to publish the peer review history of their article (what does this mean?). If published, this will include your full peer review and any attached files.

Reviewer #1: No

Reviewer #2: No

---

## [Author Response · Author response to Decision Letter 0]

13 Aug 2020

We thank the reviewers for their detailed and insightful comments, and appreciate the opportunity to improve the manuscript. Specific reviewer comments are addressed below, with line numbers referring to the final “manuscript_clean” version of the manuscript.

Reviewer #1: This is a very interesting paper that contributes to understand, using an innovative method of analysis, which are the qualitative and quantitative parameters guiding the choices of a part of American craft chocolate consumers. 

Author Response: Thank you!

However, it should be noticed that the number of consumers selected is very small (only 27 subjects) and that the group was chosen in a restricted geographical area, that cannot represent the “American consumers” in general. This aspect has been reported in the “limits of the study” section, but, for this reason, the title should be modified because it suggests a larger sample including different geographic areas. 

Author Response: Thank you for your comment. We believe that insights from these groups are generalizable to the broader American public due to the convergent validity with the FCIA and NCA studies (NCA, 2019; FCIA, 2018) lines 1089-1091. We do not believe that these results are generalizable to populations outside of the United States and therefore kept American in the title to reflect that fact. 

Furthermore, even though very appealing, the title reports the “golden chocolate coating sheet” which is referred to a specific brand in the manuscript.

Author Response: Thank you for your comment. Reviewer 2 had similar feedback as well and we have modified the title to eliminate the “golden ticket” part.

Some other aspects need modifications and/or more explanations:

Line 55: The authors reported only two trade names as an example of "gourmet" chocolate. This aspect seems to be unfair from a commercial and marketing point of view. It would be better to report a table or a link to a website (if available) where the different categories and trade names are listed. The same observation can be done for “Dandelion Chocolate,” Line 57.

Author Response: Thank you for highlighting this. We have removed all mentions of product names as suggested. This section was included to show how chocolate categories have been demarcated by the NCA in the past and now present. Because these are the chocolate products listed in the cited article (Thompson, 2016), they seemed appropriate to mention for clarity and ease of reading. However, because this was a past way to segment the category, it does not warrant a table in our opinion. If readers are interested, they are referred to the cited article or are able to look up the price of specific products to determine how much they cost per pound and what category they would fit in. 

Line 61-65: On the basis of which criteria these commercial examples have been chosen?

Author Response: Thank you for your comment. We are not entirely certain what you are referring to. We have not chosen these commercial samples. Rather, they are the commercial samples that the NCA uses in their definition of “mainstream,” “premium,” and “fine chocolate.” See reference (NCA, 2019). 

Line 201: The consumption of chocolate "two to three times per month” doesn’t be considered a frequent consumption. Please explain.

Author Response: The purpose of our study was to understand what attributes are desirable for and how craft chocolate is perceived by engaged premium chocolate consumers rather than the ends of the spectrum, which are “heavy users” or “infrequent users.” Mintel, a well-respected market research company, defines a “chocolate consumer” as “18+ and purchased chocolate within the last three months” (Mintel, 2018). Unfortunately, there is no definition for an “engaged premium chocolate consumer” or even a “premium chocolate consumer” and therefore we had to create our own criteria using information from the available market research and best practices according to Stone, Bleibaum, & Sidel, 2012. We believe that someone who purchases and eats chocolate from the brands we listed “two to three times per month” qualifies as an engaged premium chocolate consumer. We have updated the text to explain this better. See lines 196-204.

Line 202-203: See comments reported above for the statements at Line 55.

Author Response: Lines 202-203 state: “frequent chocolate consumption (from daily to two to three times per month); weekly to monthly consumption of premium chocolate (Godiva, Lindt, Guittard, Eclat, Dandelion, Ghirardelli, Vosges, etc.).” The products listed are example chocolate products that we selected from the 47 samples used in the projective mapping exercise to give consumers an idea of premium or craft chocolates in an effort to help consumers put themselves in the correct bucket in the screener. It is unlikely that a consumer would identify as a “premium chocolate consumer” outright and therefore these chocolates were selected at random from the list of premium and craft chocolates used in the projective mapping activity. 

Line 204: How many subjects have been initially screened? Please add this information in the text.

Author Response: Thank you for this comment. The text has been updated to reflect the initial number of 625 screened participants. See line 204.

Line 248: Please explain with more details why these brands were chosen in the study.

Author Response. Thank you for your comment. Please see lines 249-262 for more information. The brands chosen are all popular, readily available brands in the American chocolate market, that possessed attributes we wanted to investigate. 

Line 308: Please briefly explain the "scissor-and-sort" approach.

Author Response. Thank you for your comment. Lines 296-298 have been reworked to state the following: The “classic approach” otherwise known as the “scissor-and-sort” technique, was used to cut up the printed transcripts, group similar quotes, and then assign the quotes to codes [50,55–57].

The “scissor-and-sort” or “classic” approach is another way of saying that the transcripts were coded using scissors to cut up and then sort the quotes into codes. We have referenced several sources that describe step-by-step how to code in this method and believe that this provides sufficient detail to the readers (Charmaz, 2006; Stewart et al., 2007; Krueger, 1998; Krueger & Casey, 2009).

Line 321: The description of participants’ reaction is reported only for Dandelion chocolate; it should be described for the other categories as well.

Author Response: Thank you for your comment. For all of the other chocolate bars participants were excited to eat chocolate. However, for the Dandelion chocolate bar, there was additional excitement for that specific product, this excitement was perceived as a quality determinant, and this is why it was discussed. This emotional reaction is elicited in qualitative work and therefore worth remarking on and interpreting. The text in lines 330-332 has been reworded to demonstrate the contrast in reactions by the participants to the other chocolate products. 

Line 454-455: The specific marketplace names cited by a participant should not be reported as have been correlated with chocolate quality.

Author Response: The theme emerged that where chocolate was available was a proxy for quality. We have updated the quotation to reflect more generic store types instead of names. Please see lines 469-474.

Fig.3 should be integrated with the attributes used by consumers to define the quality "Research question 2".

Author Response: This is an excellent idea. The research questions have been reworded into objectives in lines 80-82 so that the flow is improved and the results section makes more sense. Upon suggestion of reviewer 2, the search, credence, and experience cues suggested by Darby & Karni, 1973 have been used to organize the text and also incorporated into the figure. Please see revised figure 3, which is now figure 2.

Minor typing errors: please correct "two listserves" at line 195.

Author Response: This has been corrected to reflect “two electronic mailing lists” in lines 192-194. 

 

Reviewer #2: General comments

The structure of the paper should be improved. The text is quite long and readability would improve from writing more concisely. I would also strongly recommend to include more subtitles and to focus on highlighting clear take-away messages for the reader. For example, the section on ‘packaging’ is six pages of continuous text (p26-32), while the section on ‘trust’ is even double that (p32-p43). Subtitles are needed.

Author Response: Thank you for this excellent comment. We agree that the manuscript is long. We had an unexpected amount of qualitative data and tried organizing it the best way possible. We have attempted to write as concisely as possible and shortened the manuscript during revision by 700 words. We have also added additional subtitles as suggested, and believe the reworked manuscript is now easier to follow.

It would also interesting to start the result section with an overview of the main elements addressed by the participants and I think these main elements can be categorized in two (or more) categories. From my reading, it may be interesting to distinguish objective (cacao percentage, price, packaging…) vs subjective (trust, joy, perceived sustainability…) product characteristics. Another possible distinction that may help to structure the participants’ arguments is between search, experience and credence characteristics. See, for an example including an experiment with a chocolate bar: Wright, A.A., & Lynch Jr, J.G. (1995). Communication effects of advertising versus direct experience when both search and experience attributes are present. Journal of Consumer Research, 21(4), 708-718.

Author Response: This is a wonderful suggestion and the text has been updated to reflect the classification of emerging themes according to the framework of search, experience, and credence attributes (Darby & Karni, 1973).

I am especially concerned about the impact of order effects on the results. Each group was confronted with five types of chocolate in a fixed order. However, this order has (very likely) an impact on the product attributes that respondents discuss. Suppose the first type of chocolate inspires a discussion on, for example, taste and nostalgia. When the second type is presented, participants may feel that taste has already been discussed in details and thus that they should focus on the next topic e.g. the packaging. This would lead to a discussion of more obvious characteristics for the first types of chocolate and of secondary characteristics for the types presented later. It would thus be advisable to randomly vary the order of the chocolate over the focus groups.

Author Response: Thank you for your comment. This is something we discussed in length prior to conducting this research. We have added text in lines 262-270 to clarify our selection of this tasting order, which was based upon our concern for potential sensory carryover effects and fatigue due to increasing flavor complexity and intensity of the chocolates (Lawless & Heymann, 2010). We have added this as a limitation to the study in lines 1103-1108. 

I am also concerned about the impact of having one of the five chocolate bars that is ‘new’ to the participants (the Dandelion bar), while the others were already presented in the projective mapping exercise. Now an additional factor, familiar vs new, is introduced into the study which makes the interpretation of the results more challenging.

Author Response: Thank you for your comment. The Dandelion chocolate bar used in the projective mapping activity (see S1 Fig and S2 Fig) and in the focus group activities looked almost the same. The only difference was the origin name (Venezuela versus Madagascar), which is printed on a small sticker on the front of the packaging with no other differences. We couldn’t use the same bar in the focus group activity due to lack of availability. We believe that a craft chocolate bar is inherently “new” to the consumers because it isn’t readily available in supermarkets in the area and several participants admitted to not having tried it before. Therefore, while inconvenient, we feel comfortable using the Madagascar bar in the focus groups and the Venezuela bar in the projective mapping activity. We have clarified the text to explain this in more detail. Please see lines 270-272.

The impact of gender on the discussion should also be presented differently. The authors assume that there will be a difference between men and women when it comes to the appreciation of premium chocolate. However, the contribution of the study would be improved if one or two mixed (half male, half female) groups would have been included. Also, it would be interesting to comment on the presence or absence of gender effects in the projective mapping activity.

Author Response: We separated the focus groups in terms of gender because studies have revealed that gender differences exist in terms of chocolate craving due to perimenstruation in American Women. Bearing this in mind, we followed best practices for creating a comfortable focus group environment as suggested by Krueger & Casey, 2009. See lines 237-240. We did not expect the men and women to have a different appreciation of premium chocolate, but we did expect them to feel more comfortable in the discussion once separated. The result was that women in particular expressed their cravings for chocolate during perimenstruation. Future work on gender differences in desired chocolate attributes would be interesting to understand and then market to different genders, and we have added that this is a future research direction (see lines 1118-1120).

Past studies have shown that occasional and frequent users of certain products behave differently and focus on different characteristics. Why does the current study only focus on frequent users? The market share of occasional chocolate consumers may be equally larger or even larger. The focus on frequent users may also introduce a type of bias to the results. As they are selected because they are ’chocolate experts’, they may feel a drive to show off their expertise and to demonstrate the variety of their knowledge.

Author response: It would be interesting to study infrequent users, however, it is out of the scope of this study. The purpose of our study was to understand what attributes are desirable for and how craft chocolate is perceived by engaged premium chocolate consumers rather than “heavy users” or “infrequent users.” Mintel, a well-respected market research company, defines a “chocolate consumer” as “18+ and purchased chocolate within the last three months” (Mintel, 2018). Unfortunately, there is no definition for an “engaged premium chocolate consumer” and therefore we had to create our own criteria using information from the available market research and best practices according to Stone, Bleibaum, & Sidel, 2012. We believe that purchasing and eating chocolate from the brands we listed “two to three times per month” qualifies as an engaged premium chocolate consumer. That being said, we do not believe this regularity of premium chocolate purchase makes these consumers “experts” by any means. We have updated the text to explain this better. See lines 196-204.

Detailed comments

- I would recommend to shorten the title as it is too long, in my opinion. A title such as “Understanding American premium chocolate consumer perception of craft chocolate and chocolate quality” is more than adequate.

Author response: Thank you for this comment. Reviewer 1 had similar sentiments. We have shortened the title as suggested.

- Abstract – Line 489: What is a ‘European aesthetic’? How does it differ from an ‘American aesthetic’?

Author response: This was a great catch. Upon further reflection, the theme was European flavor, taste, and mouthfeel rather than package visuals or aesthetics. The section has been corrected to “Chocolate or cocoa origin” and the text has been revised in lines 505-526.

- Mention in the introduction that the study deals with the US market.

Author response: This was an oversight on our part. We have added “American” to the first sentence and throughout the introduction. See lines 28-29.

- Line 33: 129 craft chocolate makers in the US? In the world?

Author response: Thank you for this comment. This has been updated to reflect the US chocolate market. See line 32.

- Line 45: why is craft chocolate dependent on time? Why only after 1996?

Author response: The American craft chocolate movement, much like the American craft beer movement, is considered something different than previous American chocolate production periods (the turn of the 20th century with Hershey and Mars) and according to several sources it started in 1996 with ScharffenBerger Chocolate Maker. It has not been discussed frequently in the literature; however, we cite the three available references: Giller, 2017; Leissle, 2017; Woolley et. al, submitted. Lines 29-33.

- Line 52: change verb tense in text: the present tense is used while the text discusses old definitions that have been adapted.

Author response: The tense has been changed.

- Line 173: please rephrase – focus groups are not ‘the only suitable method’. A series of non-standardize in-depth interviews may also be a valid approach.

Author response: We have deleted that part of the sentence and stated that it is “a” suitable method.

- Line 183: mention that the mapping of chocolate relates to quality. Motivate this choice.

Author response: Thank you for your comment. Our instructions for the mapping activity prompted participants to group by quality as we thought this would be universally understood without priming them too much. We have updated the text to include examples of the types of relationships that the products could be mapped by See lines 178-180. Additionally, we have justified why the word “quality” was used rather than “desirable attributes” or otherwise. See lines 222-225.

- Line 263: which version was used in the study? Why was the label changed? How?

Author response: Thank you for your comment. This text has been updated for clarification. See lines 270-272 and S1 Fig and S2 Fig.

- Line 271-272: the text mention that four focus groups made ‘it possible to reach the point of theoretical saturation’. Did this in fact happen in the current study?

Author response: We have updated the text to read that “we identified key themes within financial constraints.” See lines 277-278.

- Line 291: explain first why only 24 maps were used and not 27.

Author response: The text has been rearranged so that the flow is clearer and reason for the unusable maps is explained. See lines 307-308.

- Line 293: why were three maps ‘unusable’?

Author response: The text has been updated to reflect exactly why the maps were unusable, which was that three participants did not follow instructions. See lines 307-308. 

- Line 334: the reference to “T.J.” “Maxx” was not clear to me. As a European I am not familiar with American stores. Maybe explain briefly.

Author response: Upon further inspection, T.J. Maxx was not seen as critical to the results and is reflected later in the “availability” section. Therefore, it has been removed. 

- Line 337: the mention of the name “Greg” was very puzzling at this point. Consider reframing.

Author response: We have added more text to contextualize the name “greg” and further text to clarify what will happen with these words later in the text. See lines 351-353.

- Line 429: the observation that higher prices are associated with higher quality levels off for wine. So maybe for chocolate as well.

Author response: We have added the phrase “including wine” as well as several references that demonstrate the price-quality relationship in wine, including Veale & Quester, 2009; Hall & Lockshin, 2000; Charters & Pettigrew, 2007; Ginon et al. 2014. See lines 448-449. 

- Line 566: the link between color cold and expensive, high-quality product is very likely to be context dependent (product, cultural, time…) and may be influenced by the biases in this study. Thus this type of statements should be phrased carefully.

Author response: This is an excellent comment. We have changed the text accordingly. See lines 581-597.

- Line 698: not sure if this study can be labeled as a ‘forced’ DCE since an opt-out was included.

Author response: The text has been updated to reflect that it was a “choice experiment” rather than a “forced” DCE. See lines 676-678.

- Line 725: Link to greenwashing may be made here

Author response: Thank you for your comment. We have incorporated it into the manuscript. See line 731.

- Line 750: Why not start a new section here? Because it reads a discussion.

Author response: We have added the other subtitles and clarified that results will be presented followed by a discussion. We believe this should be clear to the reader.

- Line 1149: DCE still suffer from attitude-behavior gap issues - especially price is not considered correctly by respondents, but also product availability and the selection of attributes in the study matter.

Author response: This is an excellent point and we have changed the text to reflect that it might better flesh out willingness to buy or willingness to pay, but not necessarily fix the attitude-behavior gap. See lines 1120-1125.

---

## [Decision Letter · Decision Letter 1]

22 Sep 2020

Understanding American premium chocolate consumer perception of craft chocolate and desirable product attributes using focus groups and projective mapping

PONE-D-20-12240R1

Dear Dr. Helene Hopfer

We’re pleased to inform you that your manuscript has been judged scientifically suitable for publication and will be formally accepted for publication once it meets all outstanding technical requirements.

Kind regards,

Patrizia Restani, Ph.D.

Academic Editor

PLOS ONE

Additional Editor Comments (optional):

The authors took into account the reviewers' comments and modified the text satisfactorily

Reviewers' comments:

Reviewer's Responses to Questions

**Comments to the Author**

1. If the authors have adequately addressed your comments raised in a previous round of review and you feel that this manuscript is now acceptable for publication, you may indicate that here to bypass the “Comments to the Author” section, enter your conflict of interest statement in the “Confidential to Editor” section, and submit your "Accept" recommendation.

Reviewer #1: All comments have been addressed

Reviewer #2: All comments have been addressed

2. Is the manuscript technically sound, and do the data support the conclusions?

Reviewer #1: Partly

Reviewer #2: Yes

3. Has the statistical analysis been performed appropriately and rigorously? 

Reviewer #1: Yes

Reviewer #2: N/A

4. Have the authors made all data underlying the findings in their manuscript fully available?

Reviewer #1: No

Reviewer #2: Yes

5. Is the manuscript presented in an intelligible fashion and written in standard English?

Reviewer #1: Yes

Reviewer #2: Yes

6. Review Comments to the Author

Reviewer #1: The manuscript can be accepted for publication even if the very small number of participants is a critical point that affects the conclusion made by the authors. However, this aspect has been reported in the "study limitations" section. For this reason, I suggest to change the manuscript title in "Understanding premium chocolate consumer perception of craft chocolate and desirable product attributes in Pennsylvania using focus groups and projective mapping" or in a similar title suggesting that the study is referred to a restrict american area.

Reviewer #2: (No Response)

7. PLOS authors have the option to publish the peer review history of their article (what does this mean?). If published, this will include your full peer review and any attached files.

Reviewer #1: No

Reviewer #2: No

---

## [Editor Report · Acceptance letter]

9 Oct 2020

PONE-D-20-12240R1 

Understanding American premium chocolate consumer perception of craft chocolate and desirable product attributes using focus groups and projective mapping 

Dear Dr. Hopfer:

I'm pleased to inform you that your manuscript has been deemed suitable for publication in PLOS ONE. Congratulations! Your manuscript is now with our production department. 

Kind regards, 

on behalf of

Professor Patrizia Restani 

Academic Editor

PLOS ONE